# Evolutionary routes to biochemical innovation revealed by integrative analysis of a plant-defense related specialized metabolic pathway

Gaurav D Moghe[1†], Bryan J Leong[1,2], Steven M Hurney[1,3], A Daniel Jones[1,3], Robert L Last[1,2]*

[1]Department of Biochemistry and Molecular Biology, Michigan State University, East Lansing, United States; [2]Department of Plant Biology, Michigan State University, East Lansing, United States; [3]Department of Chemistry, Michigan State University, East Lansing, United States

**Abstract** The diversity of life on Earth is a result of continual innovations in molecular networks influencing morphology and physiology. Plant specialized metabolism produces hundreds of thousands of compounds, offering striking examples of these innovations. To understand how this novelty is generated, we investigated the evolution of the Solanaceae family-specific, trichome-localized acylsugar biosynthetic pathway using a combination of mass spectrometry, RNA-seq, enzyme assays, RNAi and phylogenomics in different non-model species. Our results reveal hundreds of acylsugars produced across the Solanaceae family and even within a single plant, built on simple sugar cores. The relatively short biosynthetic pathway experienced repeated cycles of innovation over the last 100 million years that include gene duplication and divergence, gene loss, evolution of substrate preference and promiscuity. This study provides mechanistic insights into the emergence of plant chemical novelty, and offers a template for investigating the ~300,000 non-model plant species that remain underexplored.
DOI: https://doi.org/10.7554/eLife.28468.001

*For correspondence:
lastr@msu.edu

Present address: [†]Section of Plant Biology, School of Integrative Plant Sciences, Cornell University, Ithaca, United States

Competing interests: The authors declare that no competing interests exist.

## Introduction

Since the first proto-life forms arose on our planet some four billion years ago, the forces of mutation, selection and drift have generated a world of rich biological complexity. This complexity, evident at all levels of biological organization, has intrigued humans for millennia (*Tipton, 2008*; *Mayr, 1985*). Plant metabolism, estimated to produce hundreds of thousands of products with diverse structures across the plant kingdom (*Fiehn, 2002*; *Afendi et al., 2012*), provides striking examples of this complexity. Plant metabolism is traditionally divided into primary and secondary/specialized, the former referring to production of compounds essential for plant development and the latter encompassing metabolites documented as important for plant survival in nature and metabolites of as yet unknown functional significance (*Pichersky and Lewinsohn, 2011*; *Moghe and Last, 2015*). While primary metabolism generally consists of highly conserved pathways and enzymes, specialized metabolic pathways are in a state of continuous innovation (*Milo and Last, 2012*). This dynamism has produced numerous lineage-specific metabolite classes such as steroidal glycoalkaloids in Solanaceae (*Wink, 2003*), benzoxazinoid alkaloids in Poaceae (*Dutartre et al., 2012*), betalains in Caryophyllales (*Brockington et al., 2015*) and glucosinolates in Brassicales (*Halkier and Gershenzon, 2006*). The structural diversity produced by lineage-specific pathways makes them exemplary systems for understanding the evolution of novelty in the living world.

**eLife digest** There are about 300,000 species of plant on Earth, which together produce over a million different small molecules called metabolites. Plants use many of these molecules to grow, to communicate with each other or to defend themselves against pests and disease. Humans have co-opted many of the same molecules as well; for example, some are important nutrients while others are active ingredients in medicines.

Many plant metabolites are found in almost all plants, but hundreds of thousands of them are more specialized and only found in small groups of related plant species. These specialized metabolites have a wide variety of structures, and are made by different enzymes working together to carry out a series of biochemical reactions.

Acylsugars are an example of a group of specialized metabolites with particularly diverse structures. These small molecules are restricted to plants in the Solanaceae family, which includes tomato and tobacco plants. Moghe et al. have now focused on acylsugars to better understand how plants produce the large diversity of chemical structures found in specialized metabolites, and how these processes have evolved over time.

An analysis of over 35 plant species from across the Solanaceae family revealed hundreds of acylsugars, with some plants accumulating 300 or more different types of these specialized metabolites. Moghe et al. then looked at the enzymes that make acylsugars from a poorly studied flowering plant called *Salpiglossis sinuata*, partly because it produces a large diversity of these small molecules and partly because it sits in a unique position in the Solanaceae family tree. The activities of the enzymes were confirmed both in test tubes and in plants. This suggested that many of the enzymes were "promiscuous", meaning that they could likely use a variety of molecules as starting points for their chemical reactions. This finding could help to explain how this plant species can make such a wide variety of acylsugars. Moghe et al. also discovered that many of the enzymes that make acylsugars are encoded by genes that were originally copies of other genes and that have subsequently evolved new activities.

Plant scientists and plant breeders value tomato plants that produce acylsugars because these natural chemicals protect against pests like whiteflies and spider mites. A clearer understanding of the diversity of acylsugars in the Solanaceae family, as well as the enzymes that make these specialized metabolites, could help efforts to breed crops that are more resistant to pests. Some of the enzymes related to those involved in acylsugar production could also help to make chemicals with pharmaceutical value. These new findings might also eventually lead to innovative ways to produce these chemicals on a large scale.

DOI: https://doi.org/10.7554/eLife.28468.002

Previous studies investigating the emergence of lineage-specific metabolite classes uncovered the central role of gene duplication and diversification in this process: for example in biosynthesis of glucosinolates (*Benderoth et al., 2006*; *Hofberger et al., 2013*; *Edger et al., 2015*), acylsugars (*Ning et al., 2015*; *Schilmiller et al., 2015*), the saponin avenacin (*Qi et al., 2004*) and various alkaloid types such as benzoxazinoid (*Frey et al., 1997*; *Dutartre et al., 2012*), acridone (*Bohlmann et al., 1996*) and pyrrolizidine (*Ober and Hartmann, 2000*; *Kaltenegger et al., 2013*). Duplications of members of enzyme families (e.g. cytochromes P450, glycosyltransferases, methyl-transferases, BAHD acyltransferases) also play major roles in generating chemical novelty, with biosynthesis of >40,000 structurally diverse terpenoids — produced partly due to genomic clustering of terpene synthases and cytochrome P450 enzymes (*Boutanaev et al., 2015*) — as an extreme example. These duplicate genes, if retained, may experience sub- or neo-functionalization via transcriptional divergence and evolution of protein-protein interactions as well as via changes in substrate preference, reaction mechanism and allosteric regulation to produce chemical novelty (*Ohno, 1970*; *Force et al., 1999*; *Moghe and Last, 2015*; *Leong and Last, 2017*). In this study, we sought to understand the molecular processes by which a novel class of plant-defense related metabolites — acylsugars — emerged and diversified in the Solanaceae family.

Acylsugars are lineage-specific plant specialized metabolites detected in multiple genera of the Solanaceae family including *Solanum* (*King et al., 1990*; *Schilmiller et al., 2010*; *Ghosh et al.,*

2014), *Petunia* (*Kroumova and Wagner, 2003*; *Liu et al., 2017*), *Datura* (*Forkner and Hare, 2000*) and *Nicotiana* (*Kroumova and Wagner, 2003*; *Kroumova et al., 2016*). These compounds, produced in the tip cell of trichomes on leaf and stem surfaces (*Schilmiller et al., 2012*; *2015*; *Ning et al., 2015*; *Fan et al., 2016a*), typically consist of a sucrose or glucose core esterified to groups derived from fatty acid or branched chain amino acid metabolism (*Figure 1A*). Despite these simple building blocks, the combinations of the sugar cores and the different acyl chains can generate diverse structures. For example, ~81 acylsugars were detected across just two accessions of the wild tomato *Solanum habrochaites* (*Ghosh et al., 2014*). Multiple studies performed under controlled lab settings (*Puterka et al., 2003*; *Simmons et al., 2004*; *Leckie et al., 2016*; *Luu et al., 2017*) or in the wild (*Weinhold and Baldwin, 2011*) demonstrated that acylsugars mediate plant-insect and plant-fungus interactions, and hence acylsugar production has been a target for tomato breeding efforts (*Rodríguez-López et al., 2012*; *Smeda et al., 2016*). Under lab conditions, the diversity of acyl chains and the sugar cores has been shown to be functionally important in deterring insects such as spider mites, thrips and whiteflies (*Puterka et al., 2003*; *Leckie et al., 2016*). In ecological settings, ants living near *Datura wrightii* plants that produced hexanoic acid-containing acylsugars were significantly more attracted to the smell of hexanoic acid, compared to ants growing near *Nicotiana attenuata* whose acylsugars lacked hexanoic acid (*Weinhold and Baldwin, 2011*). This suggests that acyl chain diversity may be of functional consequence in the wild. However, the contribution of the large number of acylsugar structural variants in plant-insect and plant-microbe interactions is still an open question.

Acylsugars, compared to other specialized metabolic classes such as alkaloids, phenylpropanoids and glucosinolates, are biosynthetically rather simple, allowing reconstruction of the pathway in vitro. Previous research from our lab showed that cultivated tomato (*Solanum lycopersicum*) and its wild relatives produce these compounds in the tip cell of the long glandular secreting trichomes using a set of enzymes called acylsugar acyltransferases (ASATs). These enzymes catalyze sequential addition of specific acyl chains to the sucrose molecule using acyl CoA donors (*Figure 1A*) (*Schilmiller et al., 2012*; *2015*; *Fan et al., 2016a*). ASATs are members of Clade III of the large and functionally diverse BAHD enzyme family (*St Pierre and Luca, 2000*; *D'Auria, 2006*) (*Figure 1—figure supplement 1*). Despite their evolutionary relatedness, *S. lycopersicum* ASATs (SlASATs) are only ~40% identical to each other at the amino acid level. ASATs exhibit different activities across wild tomatoes due to ortholog divergence, gene duplication and neo-functionalization, leading to divergence in acceptor and donor substrate repertoire of ASATs between wild tomato species (*Schilmiller et al., 2015*; *Fan et al., 2016a*). For example, a recent study showed that the acyl CoA preference of the ASAT2 enzyme orthologs in closely related *Solanum* species is influenced by a single amino acid change, resulting in accumulation of different acylsugar products (*Fan et al., 2016a*). Similarly, duplication of the ASAT3 enzyme followed by retention and duplicate gene divergence led to the emergence of different acylsugar chemotypes across different accessions of *S. habrochaites* (*Schilmiller et al., 2015*). In addition, loss of the ASAT4 enzyme activity in northern accessions of *S. habrochaites* leads to accumulation of acylsugars lacking R2 position acetylation (*Kim et al., 2012*). Changes that alter acylsugar profiles can also occur upstream of the biosynthetic pathway — duplication of the isopropylmalate synthase enzyme involved in amino acid biosynthesis and divergence of the duplicate was shown to alter the acyl chain composition in different accessions of *S. pennellii*, presumably due to changes in the acyl CoA pools in trichomes (*Ning et al., 2015*). These evolutionary mechanisms that contribute to emergence of novel acylsugar phenotypes were studied in closely related wild tomato species and demonstrate the plasticity of the acylsugar biosynthesis in this clade of *Solanum*. Acylsugar biosynthesis and diversity, although studied very recently in *Petunia axillaris* (Petunia) (*Liu et al., 2017*), is relatively underexplored in the broader Solanaceae family, prompting the question of how the pathway evolved over a much longer time period.

In this study, we sought to understand the timeline for emergence of the ancestral ASAT activities and to explore the evolution of the acylsugar biosynthetic pathway since the origin of the Solanaceae. Typically, significant hits obtained using BLAST searches are analyzed in a phylogenetic context to understand enzyme origins (*Frey et al., 1997*; *Ober and Hartmann, 2000*; *Qi et al., 2004*; *Benderoth et al., 2006*). However, ASAT orthologs can experience functional diversification due to single amino acid changes and/or duplication (*Schilmiller et al., 2015*; *Fan et al., 2016a*), precluding functional assignment based on sequence similarity. Thus, we inferred the origins and evolution of the pathway with a bottom-up approach; starting by assessing the diversity of acylsugar

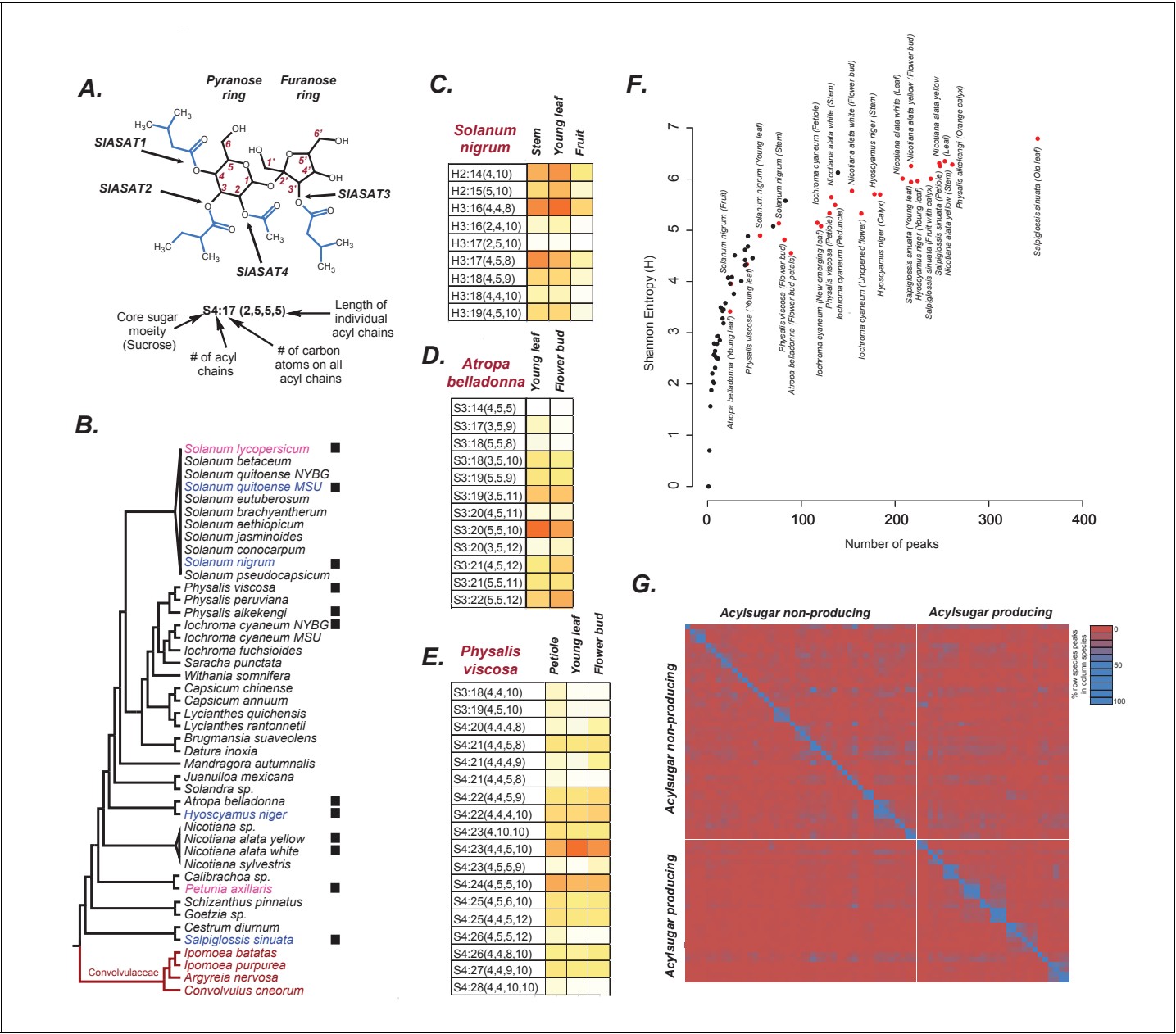

**Figure 1.** Acylsugars in solanaceae. (A) An example acylsugar from tomato. The nomenclature of acylsugars and the ASAT enzymes responsible for acylation of specific positions are described. Carbon numbering is shown in red. Sl refers to *S. lycopersicum*. The phylogenetic position of the ASAT enzymes is shown in *Figure 1—figure supplement 1*. (B) Species sampled for acylsugar extractions. Phylogeny is based on the maximum likelihood tree of 1075 species (*Särkinen et al., 2013*). Species with black squares show presence of acylsugars in mass spectrometry. Species highlighted in blue were cultivated for RNA-seq. Species in red are Convolvulaceae species. Species in pink were not sampled in this study but have been extensively studied in the context of acylsugar biosynthesis (see main text). More information about these species sampled at the NYBG is provided in *Figure 1—source data 1,2*. (C,D and E): Individual acylsugars from three representative species. Color scale ranges from no acylsugar (white) to maximum relative intensity in that species (orange). Peak areas of isomeric acylsugars were combined. *S. nigrum* produced acylsugars consistent with a hexose (H) core. Acylsugars identified from other species are described in *Figure 1—figure supplements 2* and *3*, and the raw peak intensity values obtained from different species are provided in *Figure 1—source data 2,3*. *Figure 1—figure supplement 4* shows fragmentation patterns of select acylsugars under positive ionization mode. (F) Shannon Entropy as a function of number of peaks identified. Red dots represent acylsugar producing species. Parameters used for Shannon Entropy determination and the final output are provided in *Figure 1—source data 5*. (G) Peaks shared between samples. Each row and each column represent a unique sample, with different tissues from the same species clustered together (see *Figure 1—source data 5*). Values in each cell refer to percentage of total peaks in row sample shared with the column sample. LC gradients used for all LC/MS experiments in this study are described in Figure 1—figure supplement 6.

DOI: https://doi.org/10.7554/eLife.28468.003

*Figure 1 continued on next page*

*Figure 1 continued*

The following source data and figure supplements are available for figure 1:

**Source data 1.** Species sampled at the NYBG and analysis of their trichome types.
DOI: https://doi.org/10.7554/eLife.28468.010
**Source data 2.** Values used to generate the figures in *Figure 1C–E* and *Figure 1—figure supplement 2*.
DOI: https://doi.org/10.7554/eLife.28468.011
**Source data 3.** Raw data and parameter files used to analyze Solanaceae species samples.
DOI: https://doi.org/10.7554/eLife.28468.012
**Source data 4.** mzmine 2 parameter file and peak lists.
DOI: https://doi.org/10.7554/eLife.28468.013
**Source data 5.** Data used to make *Figure 1H*.
DOI: https://doi.org/10.7554/eLife.28468.014
**Figure supplement 1.** Tomato ASATs are members of the BAHD enzyme family.
DOI: https://doi.org/10.7554/eLife.28468.004
**Figure supplement 2.** Normalized and integrated acylsugar peak areas in different species.
DOI: https://doi.org/10.7554/eLife.28468.005
**Figure supplement 3.** The complexity of acylsugar phenotype across multiple acylsugar producing species.
DOI: https://doi.org/10.7554/eLife.28468.006
**Figure supplement 4.** Positive mode CID mass spectra of select representative acylsugars from five species.
DOI: https://doi.org/10.7554/eLife.28468.007
**Figure supplement 5.** Peak and tissue specialization
DOI: https://doi.org/10.7554/eLife.28468.008
**Figure supplement 6.** Description of the LC gradients used in this study.
DOI: https://doi.org/10.7554/eLife.28468.009

phenotypes across the family using mass spectrometry. Our findings not only catalogue the diversity of acylsugars in different plants of the family but also illustrate the varied mechanisms by which the specialized metabolic pathway evolved. These results have broader implications for the study of chemical novelty in the plant kingdom.

## Results and discussion

### Diversity of acylsugar profiles across the Solanaceae

While the Solanaceae family comprises 98 genera and >2700 species (*Olmstead and Bohs, 2007*), there are extensive descriptions of acylsugar diversity reported for only a handful of species (*Severson et al., 1985*; *King et al., 1990*; *Shinozaki et al., 1991*; *Ghosh et al., 2014*). In this study, we sampled vegetative tissue surface metabolites from single plants of 35 Solanaceae and four Convolvulaceae species. These species were sampled at the New York Botanical Gardens and Michigan State University (*Figure 1B*; *Figure 1—source data 1A*), and acylsugar profiles were obtained using liquid chromatography-mass spectrometry (LC/MS) with collision-induced dissociation (CID; see Materials and methods) (*Schilmiller et al., 2010*; *Ghosh et al., 2014*; *Fan et al., 2016b*). Molecular and substructure (fragment) masses obtained by LC/MS-CID were used to annotate acylsugars in *Solanum nigrum*, *Solanum quitoense*, *Physalis alkekengi*, *Physalis viscosa*, *Iochroma cyaneum*, *Atropa belladonna*, *Nicotiana alata*, *Hyoscyamus niger* and *Salpiglossis sinuata* (Salpiglossis) (*Figure 1B,C*; *Figure 1—figure supplement 2*; *Figure 1—source data 2*). Plant extracts without detectable acylsugars generally lacked glandular trichomes (Fisher Exact Test p=2.3e-6) (*Figure 1—source data 1B*). In addition, the acylsugar phenotype is quite dynamic and can be affected by factors such as developmental stage, environmental conditions and the specific accession sampled (*Kim et al., 2012*; *Ning et al., 2015*; *Schilmiller et al., 2015*). These factors may also influence the detection of acylsugars in some species.

The suite of detected acylsugars exhibited substantial diversity, both in molecular and fragment ion masses. Most species accumulated acylsugars with mass spectra consistent with disaccharide cores — most likely sucrose — esterified with short- to medium-chain aliphatic acyl groups, similar to previously characterized acylsugars in cultivated and wild tomatoes (*Schilmiller et al., 2010*; *Ghosh et al., 2014*). However, *S. nigrum* acylsugar data suggested exclusive accumulation of

acylhexoses (*Figure 1C*), with fragmentation patterns similar to previously analyzed *S. pennellii* acyl-glucoses (*Schilmiller et al., 2012*). Mass spectra of *S. quitoense* acylsugars also revealed acylsugars with features distinct from any known acylsugars (*Figure 1—figure supplement 2*), and structures of these will be described in detail in a separate report. In total, more than 100 acylsucroses and at least 20 acylsugars of other forms were annotated with number and lengths of acyl groups based on pseudomolecular and fragment ion masses (*Figure 1—figure supplement 2*). Several hundred additional low-abundance isomers and novel acylsugars were also detected (*Figure 1—figure supplement 3*). For example, in Salpiglossis alone, >300 chromatographic peaks had *m/z* ratios and mass defects consistent with acylsugars (*Figure 1F*). This acylsugar diversity is notable when compared to ~33 detected acylsucroses in cultivated tomato (*Ghosh et al., 2014*).

Mass spectra also revealed substantial diversity in the number and lengths of acyl chains (*Figure 1C–E*, *Figure 1—figure supplement 2*). Based on negative ion mode data, the number of acyl chains on the sugar cores ranged from two to six (*Figure 1—figure supplement 2*), with chain lengths from 2 to 12 carbons. Across the Solanaceae, we found species that incorporate at least one common aliphatic acyl chain in all their major acylsugars: for example chains of length C5 in *A. belladonna*, C4 in *P. viscosa*, and C8 in *P. alkekengi* (*Figure 1D,E*; *Figure 1—figure supplement 2*). Longer C10 and C12 chain-containing acylsugars were found in multiple species (*P. viscosa*, *A. belladonna*, *H. niger*, *S. nigrum* and *S. quitoense*) (*Figure 1C–E*, *Figure 1—figure supplement 2*). Mass spectra consistent with acylsugars containing novel acyl chains were also detected. While we could not differentiate between acyl chain isomers (e.g. *iso*-C5 [iC5] vs. *anteiso*-C5 [aiC5]) based on CID fragmentation patterns, our data reveal large acyl chain diversity in acylsugars across the Solanaceae. Such diversity between species can result from differences in the intracellular concentrations of acyl CoA pools or divergent substrate specificities of individual ASATs. A previous study from our lab identified allelic variation in the enzyme isopropylmalate synthase 3, which contributes to differences in abundances of iC5 or iC4 chains in acylsugars of *S. lycopersicum* and some *S. pennellii* accessions (*Ning et al., 2015*). *ASAT* gene duplication (*Schilmiller et al., 2015*), gene loss (*Kim et al., 2012*) and single residue changes (*Fan et al., 2016a*) also influence chain diversity in acylsugars, illustrating the various ways by which acylsugar phenotypes may be generated in the Solanaceae.

Published data from *S. pennellii* and *S. habrochaites* revealed differences in furanose ring acylation on acylsucroses (*Schilmiller et al., 2015*). Ring-specific acylation patterns can be evaluated using positive mode mass spectrometry and CID, which generates fragment ion masses from cleavage of the glycosidic linkage. We found furanose ring acylation in almost all tested species in Solanaceae (*Figure 1—figure supplement 4A–E*); however, the lengths of acyl chains on the ring varied. All tri-acylsucroses analyzed using positive mode CID data bore all acyl chains on one ring, likely the pyranose ring — as evidenced by neutral loss of 197 Da (hexose plus $NH_3$) from the $[M+NH_4]^+$ ion — unlike *S. lycopersicum*, which bears one acyl chain on the furanose ring. However, the substituents varied among tetra- and penta-acylsugars of different species, with some (e.g.: *N. alata*), showing up to four chains on the same ring (*Figure 1—figure supplement 4A*), and others (e.g.: *H. niger*, Salpiglossis) revealing spectra consistent with acylation on both pyranose and furanose rings (*Figure 1—figure supplement 4B,C*).

These findings illustrated that species across the family show very diverse acylsugar profiles, prompting us to quantify the overall surface metabolite diversity using Shannon Entropy (see Materials and methods). We found that acylsugar producing species had significantly higher entropies compared to non-producing species [Kolmogorov-Smirnov (KS) test p=2e-18], indicative of the qualitative and quantitative variation in acylsugar profiles in the producers (*Figure 1F*). We also found that different tissues sampled from the same species shared a substantial proportion of peaks, however, peaks were generally unique between species — a finding supported by manual annotation of acylsugars (*Figure 1G*; *Figure 1—figure supplement 2*). Acylsugar producing species, although sharing a smaller proportion of peaks, shared relatively more peaks with other acylsugar producing species than with non-producers (*Figure 1—source data 5*). Inferences derived from additional parameters, namely peak specificity and tissue specialization (*Figure 1—figure supplement 5*), were consistent with these observations suggesting a high degree of specificity of surface metabolites in Solanaceae species.

These results demonstrate the substantial acylsugar diversity across the Solanaceae family, most of which is unique to any given species. To identify the enzymes that contribute to this diversity, we

performed RNA-seq in four phylogenetically-spaced species with interesting acylsugar profiles, namely *S. nigrum*, *S. quitoense*, *H. niger* and Salpiglossis.

## Transcriptomic profiling of trichomes from multiple Solanaceae species

Our previous studies in cultivated tomato (*Schilmiller et al., 2012*; *2015*; *Ning et al., 2015*; *Fan et al., 2016a*) demonstrated that identifying genes with expression enriched in stem/petiole trichomes compared to shaved stem/petiole without trichomes is a productive way to find acylsugar biosynthetic enzymes. We sampled polyA RNA from these tissues from four species and performed de novo read assembly (*Table 1*). These assemblies were used to find transcripts preferentially expressed in the trichomes (referred to as 'trichome-high transcripts') and to develop hypotheses regarding their functions based on homology.

Overall, 1888–3547 trichome-high transcripts (22–37% of all differentially expressed transcripts) were identified across all four species (False Discovery Rate adjusted p<0.05, fold change ≥2) (*Table 1*). These transcripts were subjected to a detailed analysis including coding sequence prediction, binning into 25,838 orthologous groups, assignment of putative functions based on tomato gene annotation and Gene Ontology enrichment analysis (see Materials and methods, *Table 1*). Analysis of the enriched categories (Fisher exact test corrected p<0.05) revealed that only 20 of 70 well-supported categories (≥10 transcripts) were enriched in at least three species (*Supplementary file 1*), suggesting existence of diverse transcriptional programs in the trichomes at the time of their sampling. Almost all enriched categories were related to metabolism, protein modification or transport, with metabolism-related categories being dominant (*Supplementary file 1*). These results support the notion of trichomes as 'chemical factories' (*Schilmiller et al., 2008*) and point to the metabolic diversity that might exist in trichomes across the Solanaceae.

A major goal of this study was to define the organization of the acylsugar biosynthetic pathway at the origin of the Solanaceae, prompting us to focus on Salpiglossis, whose phylogenetic position is of special interest in inferring the ancestral state of the biosynthetic pathway. We first validated the plant under study as Salpiglossis using a phylogeny based on *ndhF* and *trnLF* sequences (*Figure 2—figure supplement 1A,B*). A previously published maximum likelihood tree of 1075 Solanaceae species suggested *Salpiglossis* as an extant species of the earliest diverging lineage in Solanaceae (*Särkinen et al., 2013*). However, some tree reconstruction approaches show *Duckeodendron* and *Schwenckia* as emerging from the earliest diverging lineages, and Salpiglossis and Petunioideae closely related to each other (*Olmstead et al., 2008*; *Särkinen et al., 2013*). Thus, our further interpretations are restricted to the last common ancestor of Salpiglossis-Petunia-Tomato (hereafter referred to as the Last Common Ancestor [LCA]) that existed ~22–28 mya. To infer the ancestral

**Table 1.** RNA-seq data statistics

| Item | S. nigrum | S. quitoense | H. niger | S. sinuata |
|---|---|---|---|---|
| Original read pairs | 81,314,841 | 85,374,110 | 86,161,659 | 80,302,734 |
| Filtered read pairs (% original) | 73346531 (90.2%) | 76734781 (89.9%) | 76819022 (89.2%) | 71129160 (88.6%) |
| Normalized read pairs (% normalized) | 17301238 (23.6%) | 15350023 (20.0%) | 20779972 (27.1%) | 17905057 (25.2%) |
| Total transcript isoforms | 160,583 | 124,958 | 189,711 | 149,136 |
| Longest isoforms (% total) | 78,020 (49%) | 72,426 (58%) | 96,379 (51%) | 77,970 (52.3%) |
| With > 10 reads | 32,105 | 32,044 | 38,252 | 32,798 |
| With predicted peptide > 50aa | 23,224 | 22,289 | 26,262 | 23,570 |
| Differentially expressed[*,†] (%>10 reads) | 10,386 (32%) | 12,194 (38%) | 9007 (24%) | 7091 (22%) |
| Trichome high[†] (% differentially expressed) | 2292 (22.1%) | 3547 (29.1%) | 3321 (36.8%) | 1888 (26.6%) |

\* Differentially expressed genes at p<0.05 (corrected for multiple testing); [2] p<0.05, fold change >2

† Some differentially expressed transcripts were confirmed by RT-PCR, as shown in *Figure 2—figure supplement 3*.

DOI: https://doi.org/10.7554/eLife.28468.015

state of the acylsugar biosynthetic pathway in the LCA, we characterized the pathway in Salpiglossis using in vitro and *in planta* approaches.

## In vitro investigation of Salpiglossis acylsugar biosynthesis

The acylsugar structural diversity and phylogenetic position of Salpiglossis led us to characterize the biosynthetic pathway of this species. NMR analysis of Salpiglossis acylsugars revealed acylation at the R2, R3, R4 positions on the pyranose ring and R1′, R3′, R6′ positions on the furanose ring (*Figure 2A*; *Figure 2—source data 1*) . The acylation positions are reminiscent of *Petunia axillaris* (Pa) acylsucroses where PaASAT1, PaASAT2, PaASAT3 and PaASAT4 acylate with aliphatic precursors at R2, R4, R3 and R6 on the six-carbon pyranose ring, respectively (*Nadakuduti et al., 2017*). Thus, we tested the hypothesis that PaASAT1,2,3 orthologs in Salpiglossis function as SsASAT1,2,3 respectively.

Thirteen Salpiglossis trichome-high BAHD family members were found (*Figure 2—figure supplement 1C*), with nine expressed in *Escherichia coli* (*Figure 2—figure supplement 2*; *Supplementary file 2*). Activities of the purified enzymes were tested using sucrose or partially acylated sucroses (*Fan et al., 2016a*; *Fan et al., 2016b*) as acceptor substrates. Donor C2, aiC5 and aiC6 acyl CoA substrates were tested based on the common occurrence of these ester groups in a set of 16 Salpiglossis acylsucroses purified for NMR. Representative NMR structures that illustrate the SsASAT positional selectivity described in the results below are shown in *Figure 2A*. Four of the tested candidates catalyzed ASAT reactions (*Figure 2B–E*). In the following description, we name the enzymes based on their order of acylation in the Salpiglossis acylsugar biosynthetic pathway. A description is provided in *Figure 2—figure supplement 4* to assist in understanding the chromatograms.

*Salpiglossis sinuata* ASAT1 (SsASAT1) generated mono-acylsugars from sucrose using multiple acyl CoAs (*Figure 2B*, *Figure 2—figure supplements 4* and *5*), similar to the donor substrate diversity of SlASAT1 (*Fan et al., 2016a*) (*Figure 2—figure supplement 4*). We infer that SsASAT1 primarily acylates the R2 position on the pyranose ring. This is based on (a) the S1:6(6) negative mode CID fragmentation patterns (*Figure 2—figure supplement 5A*) and (b) comparisons of chromatographic migration of the mono-acylsucroses produced by SsASAT1, PaASAT1, which acylates sucrose at R2 and matches the major SsASAT1 product, and SlASAT1, which acylates sucrose at R4 (*Figure 2—figure supplement 5A,B*) (*Nadakuduti et al., 2017*). The Salpiglossis SsASAT1 activity is similar to *P. axillaris* PaASAT1 and unlike *S. lycopersicum* SlASAT1.

SsASAT2 was identified by testing the ability of each of the other eight cloned enzymes to acylate the mono-acylated product of SsASAT1 (S1:5 or S1:6) as acyl acceptor and using aiC5 CoA as acyl donor. SsASAT2 catalyzed the formation of di-acylated sugars, which co-eluted with the PaASAT2 product but not with the SlASAT2 product (*Figure 2B*; *Figure 2—figure supplement 5C,D*). Positive mode fragmentation suggested that the acylation occurs on the same ring as SsASAT1 acylation (*Figure 2—figure supplement 5C*). This enzyme failed to acylate sucrose (*Figure 2—figure supplement 6*), supporting its assignment as SsASAT2.

SsASAT3 (*Figure 2B*, red chromatogram) added aiC6 to the pyranose ring of the di-acylated sugar acceptor. This aiC6 acylation reaction is in concordance with the observed *in planta* acylation pattern at the R3 position — aiC6 is present at this position in the majority of *S. sinuata* acylsugars. Surprisingly, the enzyme did not use aiC5 CoA, despite the identification of S3:15(5,5,5) and likely its acylated derivates [S4:20(5,5,5,5), S4:17(2,5,5,5), S5:22(2,5,5,5,5) and S5:19(2,2,5,5,5)] from *S. sinuata* extracts. SsASAT3-dependent R3 position acylation was further confirmed by testing the tri-acylated product with PaASAT4, which acylates at the R6 position of the pyranose ring (*Figure 2—figure supplement 7*) (*Nadakuduti et al., 2017*). The successful R6 acylation by PaASAT4 is consistent with the hypothesis that SsASAT3 acylates the R3 position. Taken together, these results suggest that the first three enzymes generate acylsugars with aiC5/aiC6 at the R2 position (SsASAT1), aiC6 at the R3 position (SsASAT3) and aiC5/aiC6 at the R4 position (SsASAT2).

We could not identify the SsASAT4 enzyme(s) that performs aiC5 and C2 acylations on the R1′ and R3′ positions of tri-acylsucroses, respectively. However, we identified another enzyme, which we designate SsASAT5, that showed three activities acetylating tri-, tetra- and penta-acylsucroses (see Materials and methods). SsASAT5 can perform furanose ring acetylation on tri- (*Figure 2—figure supplement 8A,B*) and tetra-acylsucroses (*Figure 2C,D*), and both furanose as well as pyranose ring acetylation on penta-acylsucroses (*Figure 2—figure supplement 8C–F*). All of the products

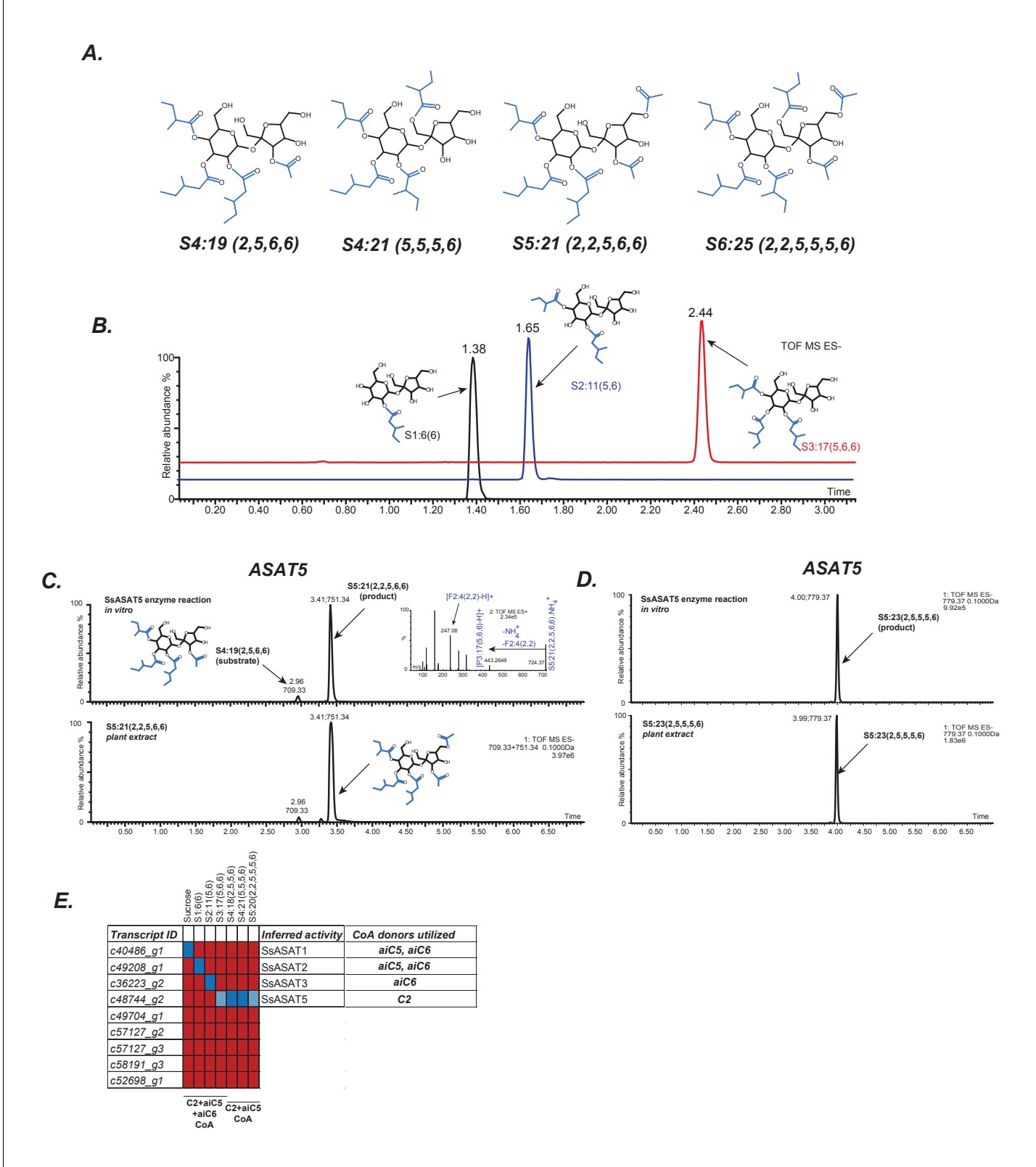

**Figure 2.** In vitro validation of Salpiglossis ASAT candidates. (A) NMR derived structures of three Salpiglossis acylsugars. NMR resonances used to interpret the first three structures are described in *Figure 2—source data 1*. We verified the plant under study as Salpiglossis using genetic markers. These results are shown in *Figure 2—figure supplement 1A,B*. (B) Results of enzyme assays for SsASAT1 (black), SsASAT2 (blue) and SsASAT3 (red). Numbers above the peaks represent the retention times of the individual compounds (see Materials and methods), whose predicted structures are

*Figure 2 continued on next page*

*Figure 2 continued*

shown alongside. Validation of trichome-high expression of candidate enzymes (*Figure 2—figure supplement 2*) is shown in *Figure 2—figure supplement 3*. Additional validation of the in vitro results is described in *Figure 2—figure supplements 4–7*. (C,D) The SsASAT5 reactions, whose products have the same retention time as *in planta* compounds. Inset in panel C shows positive mode fragmentation and predicted acyl chains on pyranose [P] and furanose [F] rings. SsASAT5 also performs additional acylation activities as shown in *Figure 2—figure supplement 8*. (E) Testing various ASAT candidates with different acceptor (top) and donor (bottom) substates. Red indicates no activity seen by LC/MS, dark blue indicates a likely true activity, which results in a product usable by the next enzyme and/or a product that co-migrates with the most abundant expected compound. Light blue color indicates that the enzyme can acylate a given substrate, but the product cannot be used by the next enzyme or does not co-migrate with the most abundant expected compound. The relationships of the enzymes with each other are shown in *Figure 2—figure supplement 1C*.

DOI: https://doi.org/10.7554/eLife.28468.016

The following source data and figure supplements are available for figure 2:

**Source data 1.** NMR chemical shifts for four acylsugars purified from Salpiglossis plants.

DOI: https://doi.org/10.7554/eLife.28468.025

**Figure supplement 1.** Phylogenetic positions of Salpiglossis, Hyoscyamus and Salpiglossis candidate enzymes.

DOI: https://doi.org/10.7554/eLife.28468.017

**Figure supplement 2.** Trichome preferentially expressed BAHD enzymes.

DOI: https://doi.org/10.7554/eLife.28468.018

**Figure supplement 3.** Confirmation of differential expression results from RNA-seq using semi-quantitative RT-PCR.

DOI: https://doi.org/10.7554/eLife.28468.019

**Figure supplement 4.** SsASAT1 reactions with different acyl CoA substrates.

DOI: https://doi.org/10.7554/eLife.28468.020

**Figure supplement 5.** Comparative analyses of LC/MS retention times of enzyme reaction products.

DOI: https://doi.org/10.7554/eLife.28468.021

**Figure supplement 6.** SsASAT2 does not acylate sucrose.

DOI: https://doi.org/10.7554/eLife.28468.022

**Figure supplement 7.** SsASAT3 acylates at the R3 position.

DOI: https://doi.org/10.7554/eLife.28468.023

**Figure supplement 8.** SsASAT5 putative secondary activities.

DOI: https://doi.org/10.7554/eLife.28468.024

produced by SsASAT5 in vitro co-migrate with acylsugars found in plant extracts, suggesting the SsASAT5 acceptor promiscuity also occurs *in planta*. Our observation that SsASAT5 can perform pyranose ring acetylation — albeit weakly (*Figure 2—figure supplement 8D,F*) — is at odds with NMR-characterized structures of a set of 16 purified Salpiglossis acylsugars, which show all acetyl groups on the furanose ring. However, a previous study described one pyranose R6-acylated penta-acyl sugar S5:22(2,2,6,6,6) in Salpiglossis (*Castillo et al., 1989*), suggesting the presence of accession-specific variation in enzyme function. Despite showing SsASAT4-, SsASAT5- and SsASAT6-like activities, we designate this enzyme SsASAT5 because its products have both acylation patterns and co-migration characteristics consistent with the most abundant penta-acylsugars from the plant (*Figure 2C,D*).

Overall, in vitro analysis revealed four enzymes that could catalyze ASAT reactions and produce compounds also detected in plant extracts (*Figure 2E*). We further verified that these enzymes are involved in acylsugar biosynthesis by testing the effects of perturbing their transcript levels using Virus Induced Gene Silencing (VIGS).

## *In planta* validation of acylsugar biosynthetic enzymes

To test the role of the in vitro identified ASATs *in planta*, we adapted a previously described tobacco rattle virus-based VIGS procedure (*Dong et al., 2007*; *Velásquez et al., 2009*) for Salpiglossis. We designed ~300 bp long gene-specific silencing constructs for transient silencing of SsASAT1, SsASAT2, SsASAT3 and SsASAT5 (*Supplementary files 2,3*), choosing regions predicted to have a low chance of reducing expression of non-target genes (see Materials and methods). The Salpiglossis ortholog of the tomato phytoene desaturase (PDS) carotenoid biosynthetic enzyme was used as positive control (*Figure 3A*), with transcript level decreases confirmed for each candidate using qRT-PCR in one of the VIGS replicates (*Figure 3B*). As no standard growth or VIGS protocol was available for Salpiglossis, we tested a variety of conditions for agro-infiltration and plant growth, and

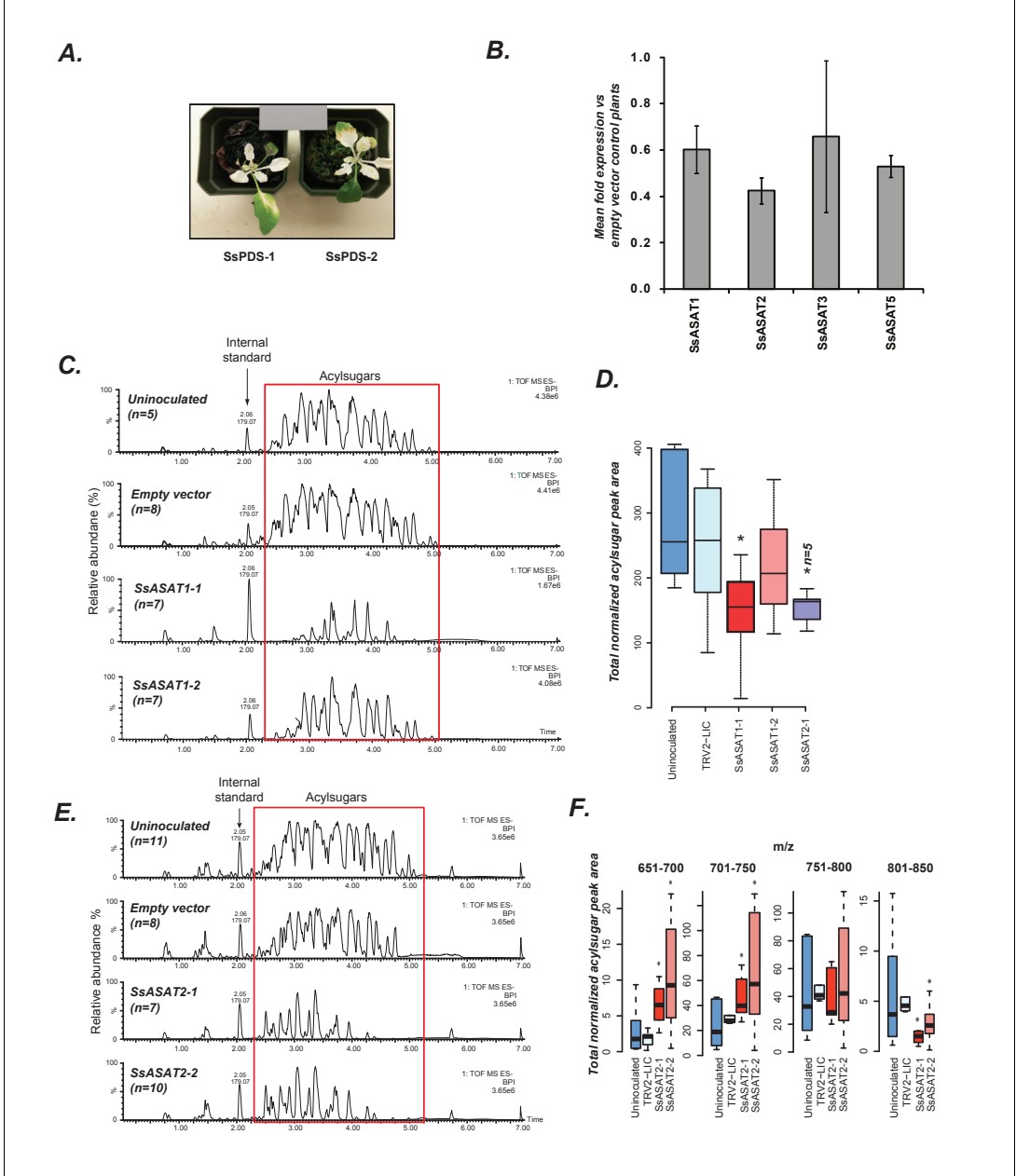

**Figure 3.** *In planta* validation of SsASAT1 and SsASAT2 candidates. (A) Two representative plants with the phytoene desaturase gene silenced using VIGS shown with 18% reflectance gray card. SsPDS-1 and SsPDS-2 have two different regions of the SsPDS transcript targeted for silencing. (B) qPCR results of SsASAT VIGS lines. Relative fold change in ASAT transcript abundance in VIGS knockdown plants compared to empty vector plants. Error bars indicate standard error obtained using three technical replicates. Expression level of the phytoene desaturase (PDS) gene was used as the reference control. *Figure 3—source data 1* includes values obtained from qPCR analysis. (C,D) SsASAT1 knockdown using two different constructs (SsASAT1-1, SsASAT1-2) shows reduction in acylsugar levels. The SsASAT1-1 phenotype is more prominent than the ASAT1-2 phenotype, being significantly lower (p=0.05; KS test). One construct for SsASAT2 (SsASAT2-1) also showed significant decrease in acylsugar levels (p=0.03; KS test). Note that the Y-axis total ion intensity in (C) is different for each chromatogram. (E,F) SsASAT2 knockdown leads to drops in levels of higher molecular weight acylsugars. In (C–F), number of plants used for statistical analysis is noted. *Figure 3—source data 2* describes normalized peak areas from VIGS plants used for making these inferences. Results of the second set of biological replicate experiments performed at a different time under a different set of conditions – as described in *Supplementary file 4* – is shown in *Figure 3—figure supplement 1*. *Figure 3—figure supplement 2* is a more detailed analysis of the SsASAT2 knockdown phenotype showing the individual acylsugar levels under the experimental conditions.
DOI: https://doi.org/10.7554/eLife.28468.026

The following source data and figure supplements are available for figure 3:

**Source data 1.** Raw and processed values obtained through qPCR.

*Figure 3 continued on next page*

*Figure 3 continued*

DOI: https://doi.org/10.7554/eLife.28468.029
**Source data 2.** Normalized peak areas calculated for VIGS experiments.
DOI: https://doi.org/10.7554/eLife.28468.030
**Figure supplement 1.** Results of knockdown of SsASAT1, SsASAT2 and SsASAT5 transcripts in a distinct replication of VIGS experiments.
DOI: https://doi.org/10.7554/eLife.28468.027
**Figure supplement 2.** Individual acylsugar levels in SsASAT2 VIGS replicate 1.
DOI: https://doi.org/10.7554/eLife.28468.028

generated at least two biological replicate experiments for each construct, run at different times (*Supplementary file 4*). ASAT knockdown phenotypes were consistent, regardless of variation in environmental conditions.

SsASAT1 VIGS revealed statistically significant reductions in acylsugar levels in at least one construct across two experimental replicates (Kolmogorov-Smirnov [KS] test, p-value=0.05) (*Figure 3C, D*; *Figure 3—figure supplement 1A*), consistent with its predicted role in catalyzing the first step in acylsugar biosynthesis. *SsASAT2* expression reduction also produced plants with an overall decrease in acylsugar levels (*Figure 3E,F*; *Figure 3—figure supplement 1B*) (KS test, p=0.03). However, these plants also showed some additional unexpected acylsugar phenotypes, namely increases in lower molecular weight acylsugars (*m/z* ratio: 651–700, 701–750; KS test p<0.05), decreases in levels of higher molecular weight products (*m/z* ratio: 751–800, 801–850) (*Figure 3E,F*) as well as changes in levels of some individual acylsugars as described in *Figure 3—figure supplement 2*. These results provide *in planta* support for the involvement of SsASAT2 in acylsugar biosynthesis, and suggest the possibility of discovering additional enzymatic activities in the future.

Silencing the *SsASAT3* transcript also led to an unexpected result - significantly higher accumulation of the normally very low abundance tri-acylsugars [S3:13(2,5,6); S3:14(2,6,6); S3:15(5,5,5); S3:16 (5,5,6); S3:18(6,6,6)] and their acetylated tetra-acylsugar derivatives (KS test p<0.05), compared to empty vector control infiltrated plants (*Figure 4A,B*; *Figure 4—figure supplement 1*). The tetra-acylsugars contained C2 or C5 acylation on the furanose ring (*Figure 4—figure supplement 2*). These observations are consistent with the hypothesis that di-acylated sugars accumulate upon SsASAT3 knockdown and then serve as substrate for one or more other enzyme (*Figure 4—figure supplement 3*). Our working hypothesis is that this inferred activity is the as-yet-unidentified SsASAT4 activity; this is based on comparisons of in vitro enzyme assay products and in vivo purified acylsugars from Salpiglossis plants (*Figure 2A*) . We propose that the hypothesized SsASAT4 may promiscuously acylate di-acylated sugars in addition to performing C2/C5 additions on tri-acylsugars.

SsASAT5, based on in vitro analysis, was proposed to catalyze acetylation of tetra- to penta-acylsugars. As expected, its knockdown led to accumulation of tri- and tetra-acylsugars (*Figure 4C,D*; *Figure 4—figure supplement 3B*). The accumulating tetra-acylsugars were C2/C5 furanose ring-acylated derivatives of the tri-acylsugars, suggesting presence of a functional SsASAT4 enzyme and further validating the annotation of the knocked down enzyme as SsASAT5 (*Figure 4—figure supplement 3B*). Thus, in summary, we identified four ASAT enzymes and validated their impact on acylsugar biosynthesis in Salpiglossis trichomes using VIGS.

Taken together, the Salpiglossis metabolites produced in vivo, combined with in vitro and RNAi results, lead to the model of the Salpiglossis acylsugar biosynthetic network shown in *Figure 5*. SsASAT1 – the first enzyme in the network – adds aiC5 or aiC6 to the sucrose R2 position. SsASAT2 then converts this mono-acylated sucrose to a di-acylated product, via addition of aiC5 or aiC6 at the R4 position. Four possible products are thus generated by the first two enzymes alone. Next, the SsASAT3 activity adds aiC6 at the R3 position, followed by one or more uncharacterized enzyme (s) that adds either C2 or aiC5 at the furanose ring R1′ or R3′ positions, respectively. SsASAT5 next performs acetylation at the R6′ position to produce penta- acylsugars, which can then be further converted by an uncharacterized SsASAT6 to hexa-acylsugars via C2 addition at the R3′ position.

Our results are consistent with the existence of at least two additional activities – SsASAT4 and SsASAT6. These enzymes may be included in the five BAHD family candidates highly expressed in both the trichome and stem (average number of reads >500), and thus not selected for our study because they did not meet the differential expression criterion. Also, the fact that there are >300 detectable acylsugar-like peaks in the Salpiglossis trichome extracts suggests the existence of

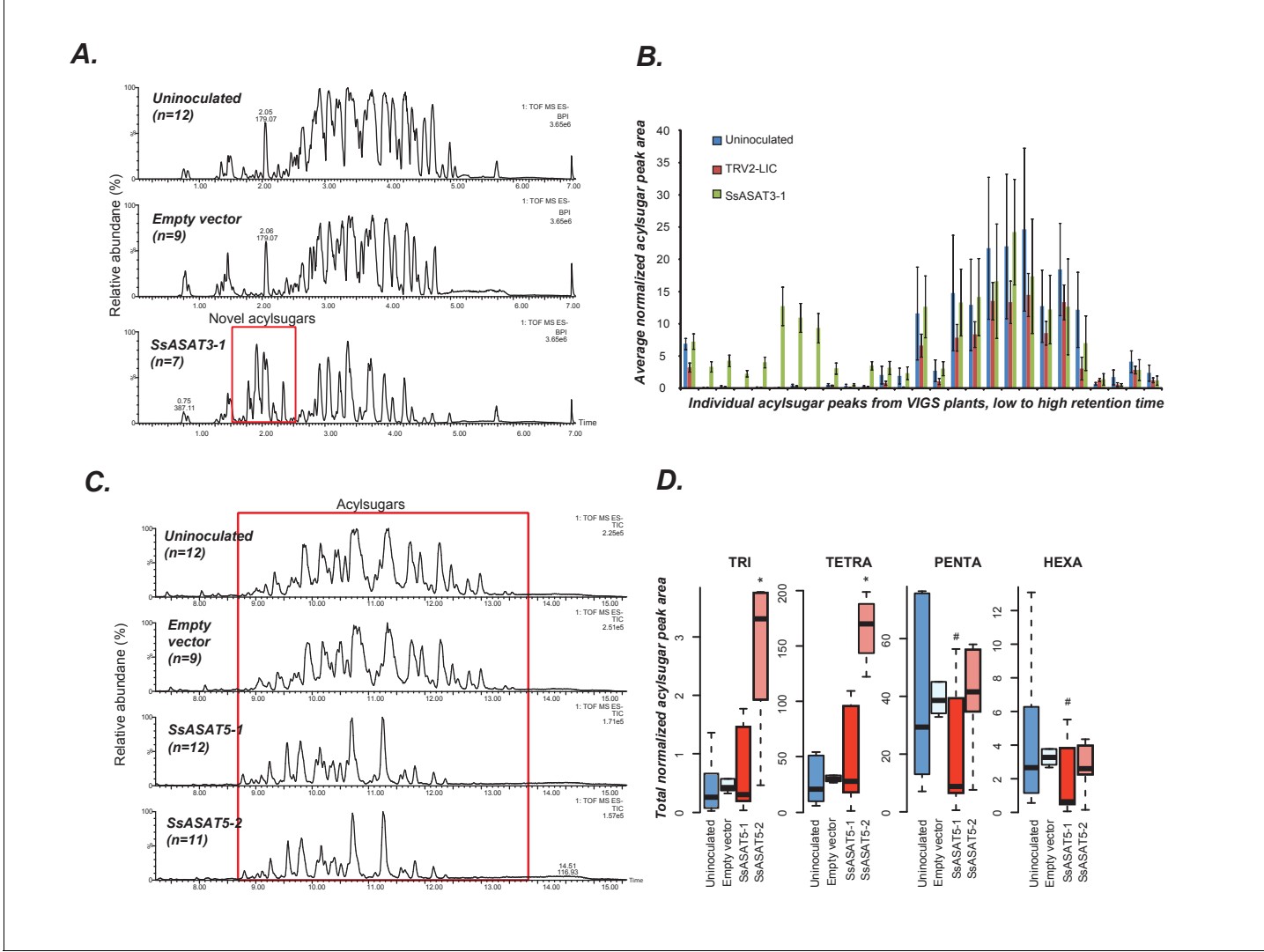

**Figure 4.** VIGS phenotypes of SsASAT3 and SsASAT5 knockdown plants. In each sub-figure, the left hand panel shows a representative chromatographic phenotype while the right hand panel shows distributions of the aggregated peak areas of all plants of the tested genotypes. (A,B) SsASAT3 VIGS knockdown experiment using a single targeting fragment SsASAT3-1 resulted in appearance of novel acylsugar peaks whose levels are significantly higher (p<0.05, KS test) vs. control. Error bars indicate standard error. Individual acylsugar peak areas are shown in *Figure 4—figure supplement 1*, while positive mode fragmentation patterns of the novel acylsugars are shown in *Figure 4—figure supplement 2*. (C) SsASAT5-1 and SsASAT5-2 chromatograms are from individual plants with two different regions of the SsASAT5 transcripts targeted for silencing. (D) Distributions of the aggregated peak areas of all plants of the tested genotypes. The boxplots show that SsASAT5 knockdown leads to a significant (*: KS test p<0.05; #: KS test 0.05 < p < 0.1) accumulation of tri-and tetra-acylsugars, and the effect is prominent in the SsASAT5-2 construct. A graphical explanation of the SsASAT3 and SsASAT5 knockdown results is presented in *Figure 4—figure supplement 3*.

DOI: https://doi.org/10.7554/eLife.28468.031

The following figure supplements are available for figure 4:

**Figure supplement 1.** SsASAT3 knockdown boxplots for levels of individual acylsugars.
DOI: https://doi.org/10.7554/eLife.28468.032
**Figure supplement 2.** Positive mode fragmentation patterns of novel acylsugars found in SsASAT3 VIGS knockdown plants.
DOI: https://doi.org/10.7554/eLife.28468.033
**Figure supplement 3.** Hypothesized routes of metabolite flow in VIGS knockdown plants.
DOI: https://doi.org/10.7554/eLife.28468.034

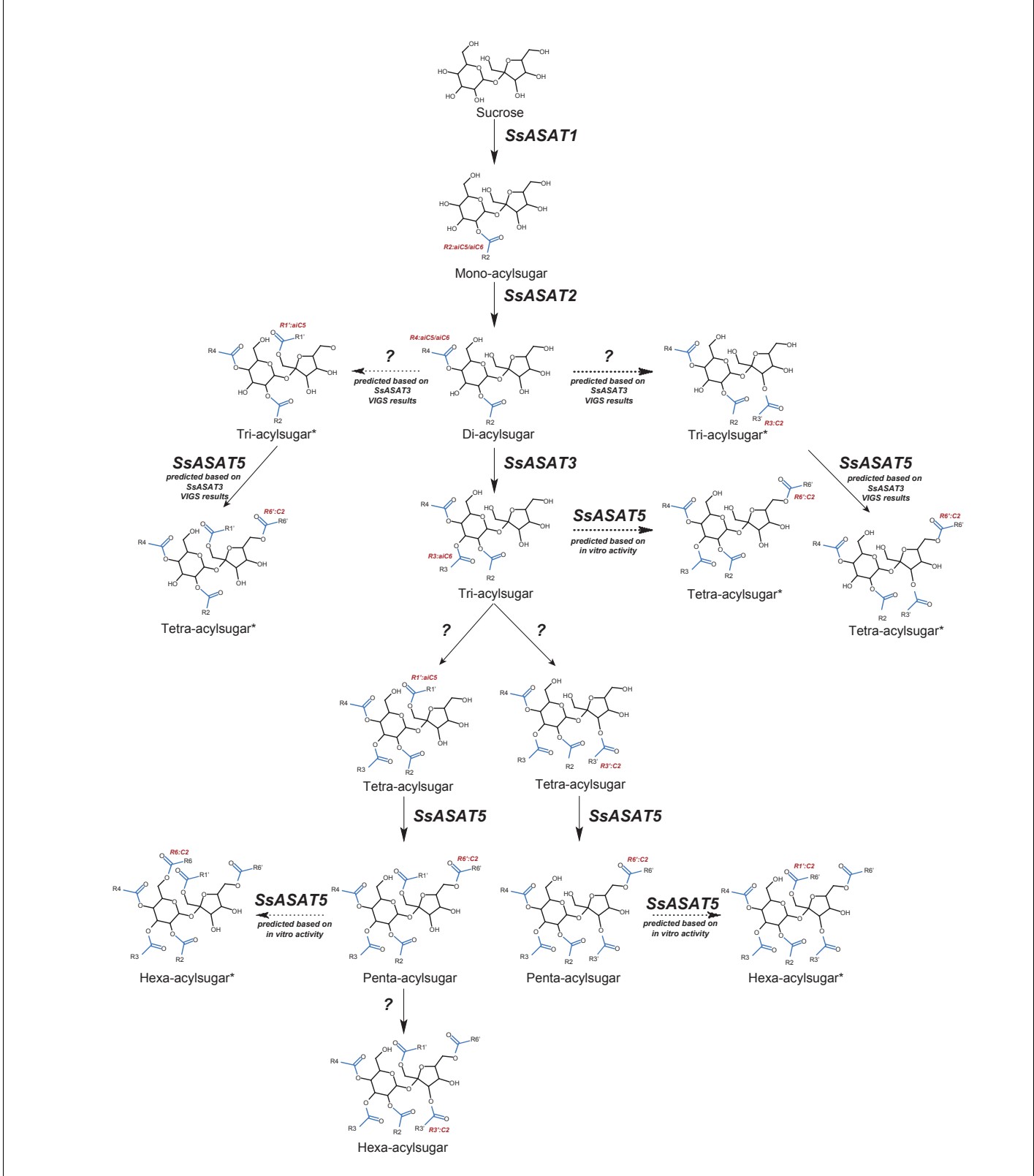

**Figure 5.** Model for the Salpiglossis acylsugar biosynthetic pathway. The question marks indicate unidentified enzymes. The blue colored acyl chains are positioned on the sucrose molecule based on results of positive mode fragmentation characteristics, co-elution assays, and comparisons with purified acylsugars. Main activities are shown in solid arrows and potential alternate activities - where acylation positions and enzymatic activities are hypothesized based on in vitro and in vivo findings - are shown in dashed arrows. An asterisk (*) next to the acylsugar names indicates no NMR

*Figure 5 continued on next page*

*Figure 5 continued*

structure is available for the acylsugar in Salpiglossis or Petunia, and the acyl chain positions are postulated based on their fragmentation patterns in positive and/or negative mode, and on hypothesized enzyme activities as described in the main text.

DOI: https://doi.org/10.7554/eLife.28468.035

additional ASAT activities, either promiscuous activities of characterized ASATs or of other uncharacterized enzymes. Nonetheless, identification of the four primary ASAT activities can help us to investigate the origins and evolution of the acylsugar biosynthetic pathway over time.

## The evolutionary origins of acylsugar biosynthesis

We used our analysis of SsASAT1, SsASAT2, SsASAT3 and SsASAT5 activities, with information about ASATs in Petunia and tomato species (*Schilmiller et al., 2012*; *2015*; *Fan et al., 2016a*; *Nadakuduti et al., 2017*), to infer the origins of the acylsugar biosynthetic pathway. Based on BLAST searches across multiple plant genomes, ASAT-like sequences are very narrowly distributed in the plant phylogeny (*Figure 6—figure supplement 1*). This led us to restrict our BLAST searches, which used SlASATs and SsASATs as query sequences, to species in the orders Solanales, Lamiales, Boraginales and Gentianales, which are all in the Lamiidae clade (*Refulio-Rodriguez and Olmstead, 2014*). Phylogenetic reconstruction was performed with the protein sequences of the most informative hits obtained in these searches to obtain a 'gene tree'. Reconciliation of this gene tree with the phylogenetic relationships between the sampled species (*Figure 6B*) allowed inference of the acylsugar biosynthetic pathway before the emergence of the Solanaceae (*Figure 6C*; *Figure 6—figure supplements 2A–C* and *3A–C*).

Three major subclades in the gene tree – highlighted in blue, red and pink – are relevant to understanding the origins of the ASATs (*Figure 6A*). A majority of characterized ASATs (blue squares in the blue subclade, *Figure 6A*) are clustered with *Capsicum* PUN1 — an enzyme involved in biosynthesis of the alkaloid capsaicin — in a monophyletic group with high bootstrap support (Group #2, red and blue subclades *Figure 6A*). Two of the most closely related non-Solanales enzymes in the tree — *Catharanthus roseus* minovincinine-19-O-acetyltransferase (MAT) and deacetylvindoline-4-O-acetyltransferase (DAT) — are also involved in alkaloid biosynthesis (*Magnotta et al., 2007*). This suggests that the blue ASAT subclade emerged from an alkaloid biosynthetic enzyme ancestor.

A second insight from the gene tree involves Salpiglossis SsASAT5 and tomato SlASAT4, which reside outside of the blue subclade. Both enzymes catalyze C2 addition on acylated sugar substrates in downstream reactions of their respective networks. Multiple enzymes in this region of the phylogenetic tree (*Figure 2—figure supplement 1C*; light blue clade) are involved in *O*-acetylation of diverse substrates for example indole alkaloid 16-epivellosimine (*Bayer et al., 2004*), the phenylpropanoid benzyl alcohol (*D'Auria et al., 2002*) and the terpene geraniol (*Shalit et al., 2003*). This observation is consistent with the hypothesis that *O*-acetylation activity was present in ancestral enzymes within this region of the phylogenetic tree.

Gene tree reconciliation with known relationships between plant families and orders (*Figure 6B*) was used to infer acylsugar pathway evolution in the context of plant evolution. We used historical dates as described by Särkinen and co-workers (*Särkinen et al., 2013*) in our interpretations, as opposed to a recent study that described a much earlier origination time for the Solanaceae (*Wilf et al., 2017*). Based on known relationships, Convolvulaceae is the closest sister family to Solanaceae; however, we found no putative ASAT orthologs in any searched Convolvulaceae species. The closest Convolvulaceae homologs were found in the *Ipomoea trifida* genome (*Hirakawa et al., 2015*) in the red subclade. This suggests that the blue and red subclades arose via a duplication event before the Solanaceae-Convolvulaceae split, estimated to be ~50–65 mya (*Särkinen et al., 2013*). Thus, this duplication event predates the whole genome triplication (WGT) event ancestral to all Solanaceae that occurred after the Solanaceae-Convolvulaceae divergence (*Bombarely et al., 2016*).

This inference is also consistent with our findings based on synonymous substitution rate (dS) distributions of homologs between cultivated tomato and Petunia. Specifically, we identified all orthologs and paralogs in the two species and obtained a distribution of all dS values (black histogram/

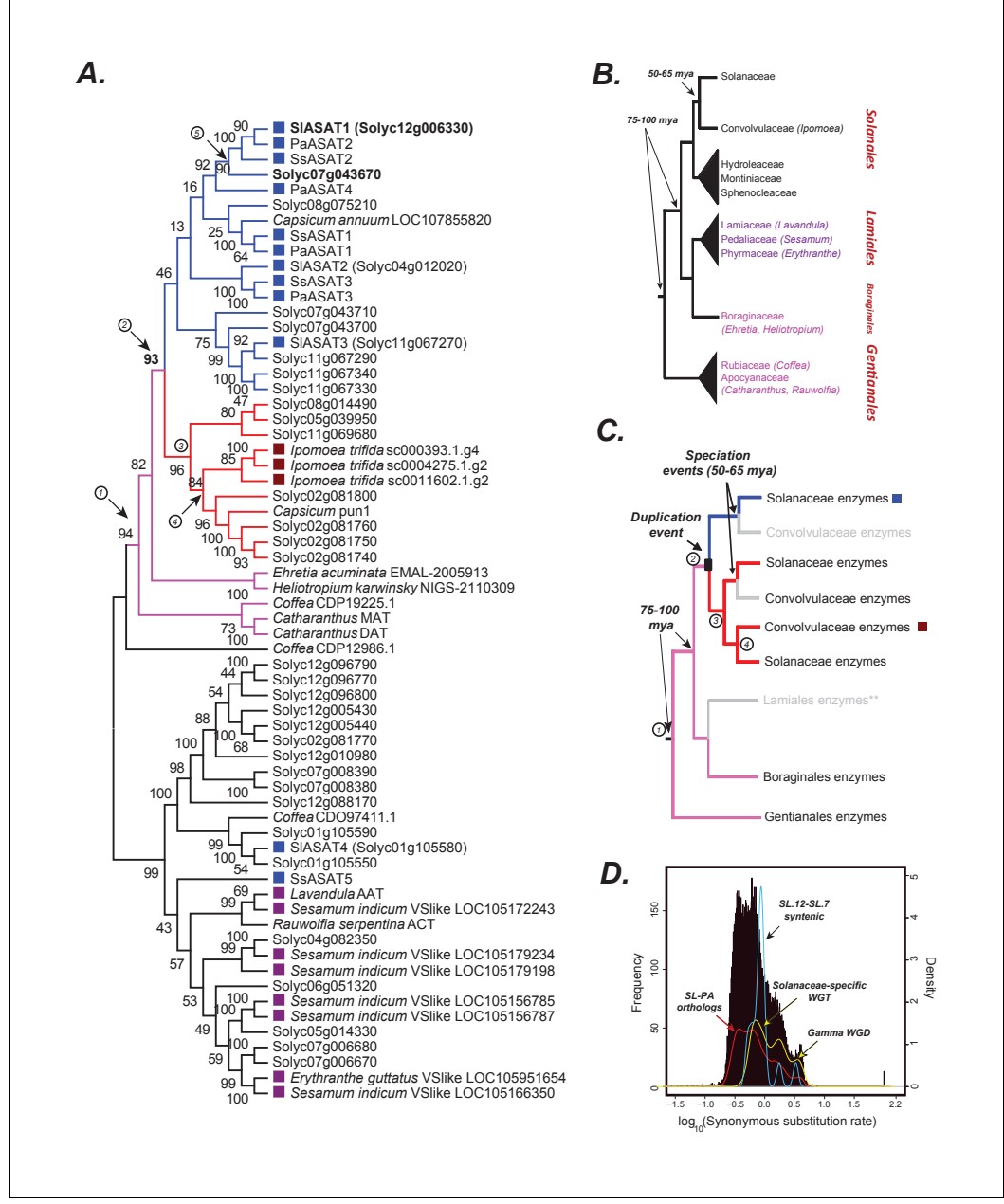

**Figure 6.** The origins of acylsugar biosynthesis. (**A**) Gene tree showing the characterized ASATs (blue squares) and other BAHDs in Clade III of the BAHD family of enzymes (*D'Auria, 2006*). Only Lamiid species are included in this tree, given significantly similar ASAT sequences were not detected elsewhere in the plant kingdom, as shown in *Figure 6—figure supplement 1*. The blue sub-clade is where most ASAT activities lie, while the red monophyletic sub-clade with high bootstrap support does not contain ASAT activities. Pink sub-clade includes enzymes present in outgroup species beyond Solanales. The purple squares highlight the closest homologs in the Lamiales order. Numbers in circles are used to denote important evolutionary events and track with the circled numbers used in (panel C), except for (5), which refers to the whole genome duplication event (panel D) that occurred prior to Solanaceae emergence. The robustness of this topology was also tested using additional tree reconstruction approaches showed in *Figure 6—figure supplements 2* and *3*. Alignment file in the MEGA mas format used for making this tree is provided in *Supplementary file 5*. *Figure 6—source data 1* shows the results of BLAST against the 1kp database. (**B**) Known relationships between different families and orders included in the study, based on *Refulio-Rodriguez and Olmstead (2014)*. The times indicated are based on a range of studies as described in the main text. (**C**) Reconciled evolutionary history of the blue ASAT sub-clade based on (panels A and B). Grey color indicates loss or lack of BLAST hits in the analyzed sequence datasets from that lineage. Species in this clade also do not produce acylsugars, as shown in *Figure 6—figure supplement 4*. ** is shown next to

*Figure 6 continued on next page*

*Figure 6 continued*

Lamiales to state the uncertainty in this inference given some uncertainty in the relationships between Lamiidae orders. Our inference is based on the strongly supported relationships in **Refulio-Rodriguez and Olmstead (2014)**(D) Synonymous substitution rate (dS) distribution of all *Petunia axillaris* (PA) - *Solanum lycopersicum* (SL) homologs. The red line plot shows the density of the histogram. The green curve shows the density of all SL-SL gene pairs in syntenic blocks. The blue curve is derived from dS of only those SL-SL gene pairs in the syntenic block encompassing SlASAT1 and Solyc07g043670, showing most gene pairs were derived using the Solanaceae-specific WGT event.

DOI: https://doi.org/10.7554/eLife.28468.036

The following source data and figure supplements are available for figure 6:

**Source data 1.** Results of BLAST against 1kp database.
DOI: https://doi.org/10.7554/eLife.28468.041
**Figure supplement 1.** The phylogenetic context of tomato ASATs.
DOI: https://doi.org/10.7554/eLife.28468.037
**Figure supplement 2.** Robustness of the phylogenetic relationships.
DOI: https://doi.org/10.7554/eLife.28468.038
**Figure supplement 3.** Additional related sequences do not affect our inferences regarding ASAT clade emergence.
DOI: https://doi.org/10.7554/eLife.28468.039
**Figure supplement 4.** LC/MS profiles of leaf surface metabolites from 11 species from the Gentianales, Lamiales, Boraginales orders collected from the living collection at the MSU Botanical Gardens.
DOI: https://doi.org/10.7554/eLife.28468.040

red curve, *Figure 6D*), as in a previously published study (*Bombarely et al., 2016*). We then specifically identified the syntenic paralogs in tomato — likely derived from the polyploidization events — and overlaid their dS values on the previous distribution (yellow curve, *Figure 6D*). This analysis differentiated the paralogs derived via ancestral WGD events from those derived via the Solanaceae-specific WGT event (*Bombarely et al., 2016*). Two BAHDs — SlASAT1 (Solyc12g006330) and Solyc07g043670 — in the blue subclade (bold, *Figure 6A*) were found to be whole genome duplicates of each other. These genes lie in a syntenic block that spans multiple gene pairs on chromosomes 12 and 7, respectively, a majority of which are derived from the Solanaceae-specific WGT event (blue curve, *Figure 6D*; *Figure 6—source data 1*). These observations were useful in annotating the duplication node separating SlASAT1 and Solyc07g043670 in the BAHD gene tree as the Solanaceae-specific WGD node (#5, *Figure 6A*). We can then also infer that the duplications giving rise to the different ASAT1,2,3 clades occurred prior to this Solanaceae-specific WGT event.

The lack of ASAT orthologs in Convolvulaceae could indicate that Convolvulaceae species with an orthologous acylsugar biosynthetic pathway were not sampled in our analysis. Alternatively, the absence of orthologs of the blue subclade enzymes in the Convolvulaceae, coupled with the lack of acylsugars in the sampled Convolvulaceae species (*Figure 1B*), leads us to hypothesize that ASAT orthologs were lost early in Convolvulaceae evolution.

The known phylogenetic relationships between species can help assign a maximum age for the duplication event leading to the blue and red subclades. Database searches identified homologs from species in the Boraginales (*Ehretia*, *Heliotropium*) and Gentianales (*Coffea*, *Catharanthus*) orders that clustered outside Group #2, in a monophyletic group with high bootstrap support (Group #1, pink branches, *Figure 6A*). Boraginales is sister to the Lamiales order (*Figure 6B*) (*Refulio-Rodriguez and Olmstead, 2014*), and although Boraginales putative orthologs were identified through BLAST searches, no orthologs from Lamiales species were detected. The closest Lamiales (*Sesamum*, *Lavandula*, *Mimulus*) homologs were more closely related to SsASAT5 than to the ASATs in the blue subclade (purple squares, *Figure 6A*). The most parsimonious explanation for these observations is that the duplication event that gave rise to the blue and red subclades occurred before the Solanaceae-Convolvulaceae split 50–65 mya but after the Solanales-Boraginales/Lamiales orders diverged 75–100 mya (*Hedges et al., 2006*) (*Figure 6B,C*). Consistent with this model, we did not find any acylsugar-like peaks in leaf surface extracts of eleven species in these orders (*Figure 6—figure supplement 4*).

These findings, which provide insights into the origin of acylsugar biosynthesis, can be interpreted under the three-step model of evolution of biological innovation, involving potentiation, actualization and refinement (*Blount et al., 2012*) (Figure 8). We propose that the ASAT enzymes catalyzing sucrose acylation first emerged (or actualized) between 30–80 mya. Events prior to the emergence of this activity — including the existence of alkaloid biosynthetic BAHDs and the duplication event 60–80 mya that gave rise to the blue and the red subclades — potentiated the emergence of the ASAT activities. Presumably, acylsugars provided a fitness advantage to the ancestral plants, leading to the refinement of ASAT activities over the next few million years. For acylsugar production to emerge, these steps in enzyme evolution would have been complemented by other innovations in ASAT transcriptional regulation leading to gland cell expression as well as production of precursors (i.e., acyl CoA donors and sucrose) in Type I/IV trichomes.

Overall, these observations support a view of the origins of the ASAT1,2,3 blue subclade from an alkaloid biosynthetic ancestor via a single duplication event >50 mya, prior to the establishment of the Solanaceae. This duplicate underwent further rounds of duplication prior to and after the Solanaceae-specific WGT event to generate the multiple ASATs found in the blue subclade. Thus, a logical next question is 'what was the structure of the acylsugar biosynthetic network early in the Solanaceae family evolution?' To address this, we focused on ASAT evolution across the Solanaceae family.

## Evolution of acylsugar biosynthesis in the Solanaceae

ASAT enzymes in the blue subclade represent the first three steps in the acylsugar biosynthetic pathway. The likely status of these three steps in the Salpiglossis-Petunia-Tomato last common ancestor (LCA) was investigated using the gene tree displayed in *Figure 7A*. Mapping the pathway enzymes on the gene tree suggests that the LCA likely had at least three enzymatic activities, which we refer to as ancestral ASAT1 (aASAT1), ASAT2 (aASAT2) and ASAT3 (aASAT3); these are shown in *Figure 7A* as red, dark blue and yellow squares, respectively. We further traced the evolution of the aASAT1,2,3 orthologs in the Solanaceae using existing functional data and BLAST-based searches. These results reveal that the aASAT1 ortholog (red squares, *Figure 7A*) was present until the *Capsicum-Solanum* split ~17 mya (*Särkinen et al., 2013*) and was lost in the lineage leading to *Solanum*. This loss is evident both in similarity searches and in comparisons of syntenic regions between genomes of Petunia, *Capsicum* and tomato (*Figure 7B*). On the other hand, the aASAT2 (dark blue squares, *Figure 7A*) and aASAT3 orthologs (yellow squares, *Figure 7A*) have been present in the Solanaceae species genomes at least since the last common ancestor of Salpiglossis-Petunia-Tomato~25 mya, and perhaps even in the last common ancestor of the Solanaceae family.

One inference from this analysis is that aASAT2 orthologs switched their activity from ASAT2-like acylation of mono-acylsucroses in Salpiglossis/Petunia to ASAT1-like acylation of unsubstituted sucrose in cultivated tomato (*Figure 7A*). Interestingly, despite the switch, both aASAT2 and aASAT3 orthologs in tomato continue to acylate the same pyranose-ring R4 and R3 positions, respectively (*Figure 7C*; *Figure 8*). In addition, cultivated tomato has two 'new' enzymes — SlASAT3 and SlASAT4 — which were not described in Petunia or Salpiglossis acylsugar biosynthesis.

The functional transitions of aASAT2 and aASAT3 could have occurred via (i) functional divergence between orthologs or (ii) duplication, neo-functionalization and loss of the ancestral enzyme. Counter to the second hypothesis, we found no evidence of aASAT2 ortholog duplication in the genomic datasets; however, we cannot exclude the possibility of recent polyploidy or tandem duplication in extant species producing duplicate genes with divergent functions. We explored the more parsimonious hypothesis that functional divergence between orthologs led to the aASAT2 functional switch.

Functional divergence between aASAT2 orthologs may have occurred by one of two mechanisms. One, it is possible that aASAT2 orthologs had some sucrose acylation activity prior to aASAT1 loss. Alternatively, sucrose acylation activity arose completely anew after the loss of aASAT1. We sought evidence for acceptor substrate promiscuity in extant species by characterizing the activity of additional orthologous aASAT2 enzymes from *H. niger* and *S. nigrum* (*Figure 7C*) using aiC6 and nC12 as acyl CoA donors. HnASAT2 — like SsASAT2 — only performed the ASAT2 reaction (acylation of mono-acylsucrose), without any evidence for sucrose acylation under standard testing conditions (*Figure 7C*, *Figure 7—figure supplement 1A–D*). However, both SnASAT1 and SlASAT1 could produce S1:6(6) and S1:12(12) from sucrose. These findings suggest that until the Atropina-Solaneae common ancestor, the aASAT2 ortholog still primarily conducted the ASAT2 activity (*Figure 7C*).

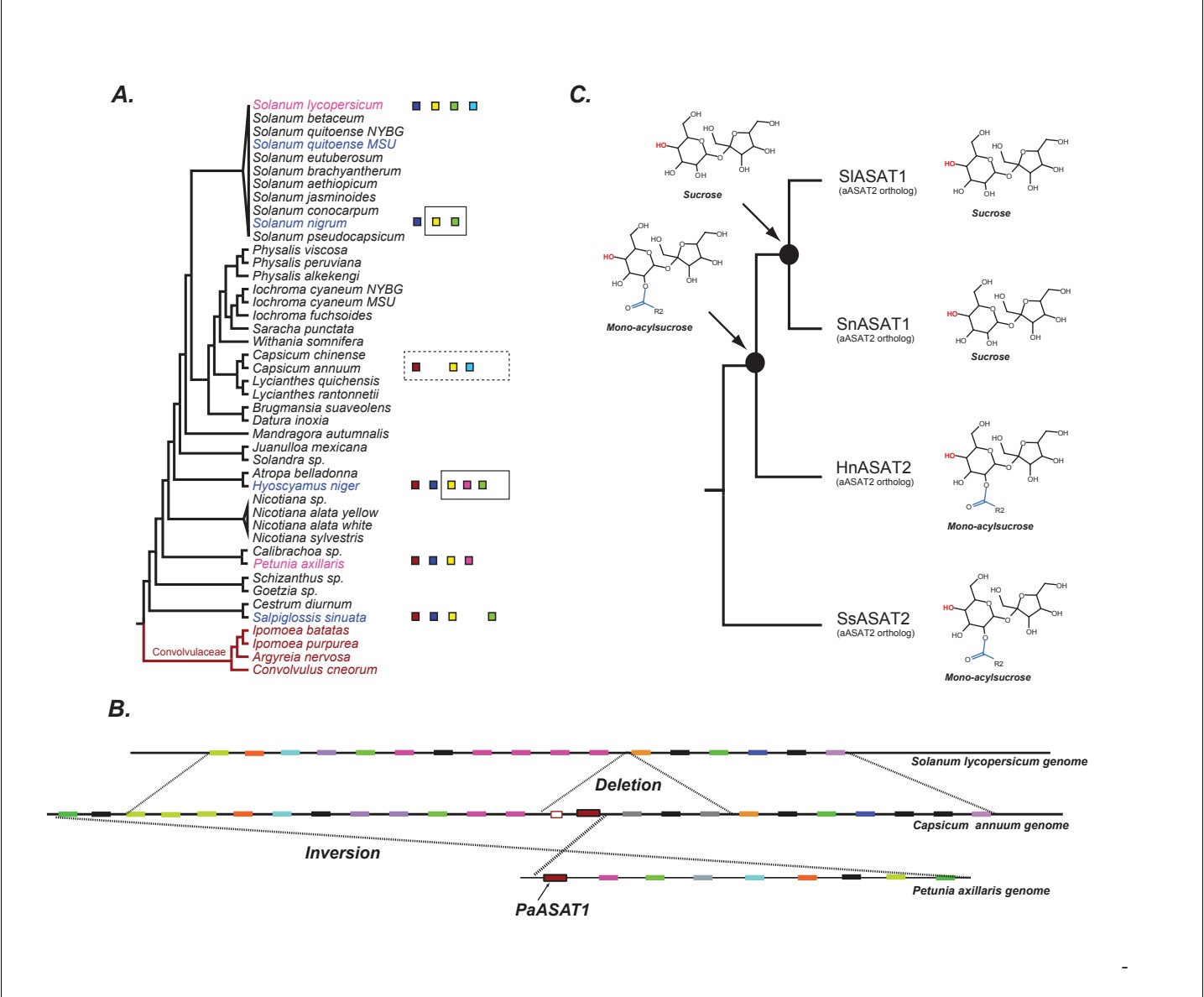

**Figure 7.** The evolution of acylsugar biosynthesis in Solanaceae.  (A) ASAT activities overlaid on the Solanaceae phylogenetic relationships. Each colored square represents a single ASAT, starting from ASAT1 and moving sequentially down the pathway to ASAT5 (left to right, sequentially). Homologs are represented by the same color. Squares not contained within a box are experimentally validated activities. A solid black box indicates that a highly identical transcript exists in the RNA-seq dataset and is trichome-high. A dashed box indicates that, based on a BLAST search, the sequence exists in the genome for the contained enzymes. In vitro validated *S. nigrum* and *H. niger* activities, including their phylogenetic positions, are highlighted in pink and Convolvulaceae species are in red. See *Figure 7—source data 1* for results of the BLAST analysis. (B) Orthologous genomic regions between three species harboring aASAT1 orthologs. Each gene in the region is shown by a colored block. Orthologous genes are represented by the same color. The PaASAT1 gene (red box) has two homologous sequences in the Capsicum syntenic region, but one of them is truncated. aASAT1 ortholog is not seen in the syntenic region in tomato. Genes used to make this figure are described in *Figure 7—source data 2*. (C) Substrate utilization of aASAT2 orthologs from multiple species is described based on the activities presented in *Figure 7—figure supplement 1*. *Hyoscyamus niger* (Hn), *Salpiglossis sinuata* (Ss), *Solanum nigrum* (Sn), *Solanum lycopersicum* (Sl). The hydroxyl group highlighted in red shows the predicted position of acylation by the respective aASAT2 ortholog.

DOI: https://doi.org/10.7554/eLife.28468.042

The following source data and figure supplement are available for figure 7:

**Source data 1.** Results of BLAST searches performed using ASAT sequences as queries and multiple databases as subjects.
DOI: https://doi.org/10.7554/eLife.28468.044
**Source data 2.** Syntenic blocks between pairs of species identified by MCScanX.
DOI: https://doi.org/10.7554/eLife.28468.045
*Figure 7 continued on next page*

*Figure 7 continued*

**Figure supplement 1.** Some aASAT2 orthologs cannot catalyze sucrose to mono-acylsucrose conversion.
DOI: https://doi.org/10.7554/eLife.28468.043

However, at some point, the aASAT2 ortholog moved in reaction space towards being ASAT1, and this activity shift was likely complete by the time the *S.nigrum-S.lycopersicum* lineage diverged (*Figure 7C*), given that aASAT2 orthologs in both species utilize sucrose.

These results point to a second round of major innovation in the acylsugar biosynthetic pathway. This round involved the following three steps: (i) Potentiation: the similarity in activities of aASAT1,2 and 3, allowing changes in substrate preferences with relatively few amino acid changes. In fact, aASAT1 and aASAT2 appear to be clustered together in the gene tree, perhaps making the aASAT2 activity switch more likely. (ii) Actualization: this refers to the first instance of the emergence of the ASAT1 activity in aASAT2 orthologs. It is unclear whether this switch occurred prior to aASAT1 loss or after. (iii) Refinement: after the actualization of the ASAT1 activity, this activity was likely refined over time to the SlASAT1 activity we see today. To understand the specific order of the events in aASAT2 evolution in the Solanaceae, ASAT1 and ASAT2 enzymes from additional species between *H. niger* and *Solanum* will need to be tested.

## Conclusions

We performed experimental and computational analysis of BAHD enzymes from several lineages spanning ~100 million years, and characterized the emergence and evolution of the acylsugar metabolic network. We identified four biosynthetic enzymes in *Salpiglossis sinuata*, the extant species of an early emerging Solanaceae lineage, characterized in vitro activities, validated *in planta* roles of these ASATs and studied their emergence over 100 million years of plant evolution. These results demonstrate the value of leveraging genomics, phylogenetics, analytical chemistry and enzymology with a mix of model and non-model organisms to understand the evolution of biological complexity.

We uncovered a large diversity of acylsugars across the family, all of which is based on two simple types of components - a sugar core (mostly sucrose) and acyl chains (C2-C12). Previous studies indicate that acylation is possible on all eight hydroxyls on the sucrose core (*Ghosh et al., 2014*; *Schilmiller et al., 2015*; *Fan et al., 2016a*). A simple calculation (see Materials and methods) considering only 12 aliphatic CoA donors typically found in Solanaceae acylsugars suggests >6000 theoretical possible structures for tetra-acylsugars alone. This estimate does not include estimates of tri-, penta-, hexa-acylsugars nor does it consider non-aliphatic CoAs such as malonyl CoA, esters of other sugars such as glucose or positional isomers. Although the theoretical possibilities are restricted by availability of CoAs in trichome tip cells and existing ASAT activities, we still observe hundreds of acylsugars across the Solanaceae, with >300 detectable acylsugar-like chromatographic peaks in single plants of Salpiglossis and *N. alata*, (*Figure 1F*) including very low abundance hepta-acylsugars and acylsugars containing phenylacetyl and tigloyl chains. The acylsugar composition is typically similar between trichomes on different plant tissues of the same individual (*Figure 1—figure supplement 2*). However, it varies at multiple taxonomic levels - that is between populations of a single species (*Kim et al., 2012*), between closely related species (*Schilmiller et al., 2015*; *Fan et al., 2016a*) and between species across the Solanaceae (this study). This structural diversity is reminiscent of other diverse lineage-specific metabolite classes such as glucosinolates in Brassicales (*Olsen et al., 2016*), betalains in Caryophyllales (*Khan and Giridhar, 2015*), acridone alkaloids in Rutaceae (*Roberts et al., 2010*), resin glycosides in Convolvulaceae (*Pereda-Miranda et al., 2010*) and pyrrolizidine alkaloids in Boraginaceae (*El-Shazly and Wink, 2014*).

Does structural diversity within the same metabolite class in a single individual provide a fitness advantage? The neutral theory of evolution serves as a null hypothesis, predicting that diversity does not confer a fitness advantage and is merely a result of drift. Indeed, the structural diversity in a single individual could be a reflection of promiscuous enzyme activities, given large enzyme families such as BAHDs, cytochrome P450s, glycosyltransferases play a role in the biosynthesis of several specialized metabolite classes. It is also possible that only a few specific metabolites belonging to the metabolite class — rather than the entire repertoire of the class — are consequential in imparting fitness advantage to an individual. Alternatively, such structural diversity may generate a meta-

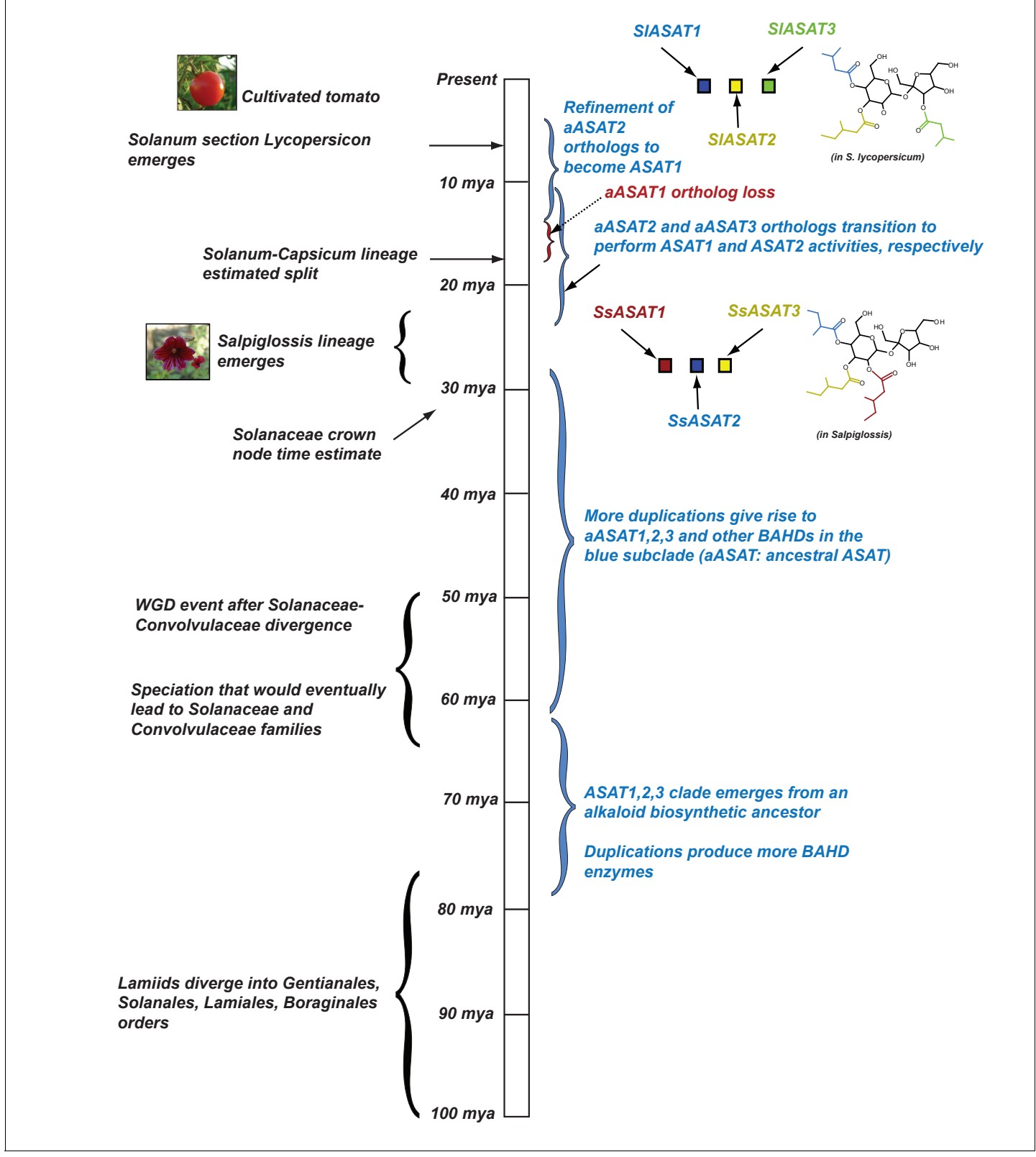

**Figure 8.** Proposed model for evolution of acylsugar biosynthesis over 100 million years. Events in plant evolution are noted on the left and the acylsugar biosynthetic pathway developments are noted on the right of the timeline. Tri-acylsugars produced by Salpiglossis and tomato ASAT1,2,3 enzymes are shown with the chain color corresponding to the color of the enzyme that adds the chain. Images obtained from Wikimedia Commons

*Figure 8 continued on next page*

*Figure 8 continued*

under Creative Commons CC BY-SA 3.0 license. We note that both Salpiglossis and Tomato ASAT activities exist in the current time, and Salpiglossis is not 'ancestral'. We use the word 'lineages' (eg: Salpiglossis lineage) to refer to the branch leading up to the currently existing (extant) species.

DOI: https://doi.org/10.7554/eLife.28468.046

property, such as stickiness — in the case of acylsugars — or synergistic antimicrobial or insect protective action that may provide a fitness advantage. A related possibility is that metabolite diversity in a single individual and, subsequently, between populations of the same species, provides the standing variation that may be acted upon by natural selection in the future, as the environment of the species changes. This was postulated to be the case for the existence of aliphatic glucosinolate diversity in Arabidopsis (*Kerwin et al., 2015*). A previous study also showed that accession-level glucosinolate diversity is maintained in European populations of Arabidopsis by the differences in relative abundance of two aphid species in different parts of the range (*Züst et al., 2012*). In the case of acylsugar diversity, we have no evidence for one or more of these hypotheses, because most studies on the importance of acylsugar structural diversity were performed in lab settings. Thus, a significant opportunity exists to characterize the importance of the structural diversity in acylsugar and other plant specialized metabolite classes in ecologically relevant settings.

Our results revealed the involvement of gene duplication and divergence, enzyme promiscuity, gene loss and functional divergence between orthologs in the evolution of the relatively compact acylsugar biosynthetic network. The process of gene loss in ASATs and other BAHDs may also have been influenced by the fractionation process occurring in the Solanaceae genomes after the Solanaceae-specific polyploidization event. Our results also point to factors such as acyl CoA pools, trichome developmental programs and genotype by environment interactions that may affect the evolution of metabolic pathways. Previous results from closely related *Solanum* species also revealed regulatory divergence of ASATs (*Kim et al., 2012*) and neo-functionalization of duplicates (*Ning et al., 2015*) in influencing the phenotypes produced by this pathway. These findings highlight the striking plasticity of the acylsugar biosynthetic network and illustrate the complexity that underlies the evolution of novel chemical phenotypes. Our results also reveal the important role played by tandem and whole genome duplications as well as gene loss in evolution of specialized metabolic pathways.

The discovery of multiple routes to biochemical innovation in our study was facilitated by the use of a highly integrative approach. Recent studies have utilized similar integrative approaches in deciphering the molecular evolution of plant-butterfly interactions in the Brassicales (*Edger et al., 2015*) and evolution of herbivore-induced defense signaling in *Nicotiana* (*Zhou et al., 2016*). Deployment of such strategies on a large scale is increasingly possible due to improvements in small molecule mass spectrometry and nucleic acid sequencing, establishment of transient knockdown protocols such as VIGS that do not require stable plant transformation, establishment of new gene editing techniques, broadly representative genome and transcriptome sequence databases and significantly better data storage and analysis abilities. Approaches such as VIGS are even more important for deciphering metabolic pathways in non-model species because of the lack of stable transformation protocols and involvement of enzyme families in specialized metabolism, which frequently requires *in planta* validation. Overall, such broad studies can provide a multi-dimensional view of the evolution of metabolite diversity because of the ability to integrate insights derived from different approaches into a single framework. In addition, phylogeny-guided studies that integrate comparative metabolomics and transcriptomics, such as co-expression analysis (*Zhou et al., 2016*; *Wisecaver et al., 2017*), can also be used for pathway discovery in non-model organisms, thus creating a robust platform for studying different aspects of biological evolution, for example co-evolutionary studies (*Edger et al., 2015*). Discovery of novel enzymatic activities, coupled with recent advances in genome editing technologies, can also benefit synthetic biology of natural products, for example as described by Ignea and co-workers (*Ignea et al., 2015*).

In this study, we primarily focused on Salpiglossis because of a unique acylsugar profile and its phylogenetic position. *S. lycopersicum* is the flagship species of the Solanaceae family, and we were able to apply the insights derived with this crop as a foundation to discover novel enzymes in related species. Our investigations in Salpiglossis were also aided by the availability of ASATs and NMR

structures of their products in Petunia (*Liu et al., 2017*; *Nadakuduti et al., 2017*), which is more closely related to Salpiglossis than tomato. 'Anchor species' such as Petunia, which are phylogenetically distant from flagship/model species, can enable study of a different region of the phylogenetic tree of the clade of interest. Development of limited genomic and functional resources in such anchor species, coupled with integrative, comparative approaches can offer more efficient routes for the exploration of biochemical complexity in the ~300,000 plant species estimated to exist on our planet (*Mora et al., 2011*).

## Materials and methods

### Plant acylsugar extractions and mass spectrometric analyses

Acylsugar extractions were carried out from plants at the New York Botanical Gardens and from other sources (*Figure 1—source data 1*). The plants sampled were at different developmental stages and were growing in different environments. The extractions were carried out using acetonitrile:isopropanol:water in a 3:3:2 proportion similar to previous descriptions (*Schilmiller et al., 2010*; *Ghosh et al., 2014*; *Fan et al., 2016a*) with the exception of gently shaking the tubes by hand for 1–2 min. All extracts were analyzed on LC/MS (Waters Corporation, USA) using 7 min, 22 min or 110 min LC gradients on Supelco Ascentis Express C18 (Sigma Aldrich, USA) or Waters BEH amide (Waters Corporation, USA) columns (Figure 1—figure supplement 6), as described previously (*Schilmiller et al., 2010*; *Ghosh et al., 2014*; *Fan et al., 2016a*). While the 110 min method was used to minimize chromatographic overlap in support of metabolite annotation in samples with diverse mixtures of acylsugars, targeted extracted ion chromatogram peak area quantification was performed using the 22 min method data. The QuanLynx function in MassLynx v4.1 (Waters Corporation, USA) was used to integrate extracted ion chromatograms for manually selected acylsugar and internal standard peaks. Variable retention time and chromatogram mass windows were used, depending on the experiment and profile complexity. Peak areas were normalized to the internal standard peak area and expressed as a proportion of mg of dry weight.

### Calculation of shannon entropy

The concept of Shannon Entropy, originally developed in the field of information theory to quantify the amount of uncertainty or information content of a message (*Shannon, 1948*), is used in ecology to quantify species diversity (*Peet, 1974*). More recently, this approach was used to quantify transcriptomic and metabolic diversity and specialization (*Martínez and Reyes-Valdés, 2008*; *Li et al., 2016*).

To calculate Shannon Entropy, we explored three different software packages for processing the RAW files from the Waters LCT Premier Mass Spectrometer, namely (i) the MarkerLynx function in MassLynx software v4.1 (Waters Corporation, USA), (ii) Progenesis QI suite (Nonlinear Dynamics, USA) and (iii) mzMine 2 (*Pluskal et al., 2010*). We found mzMine 2 most appropriate for our use, because it had several options for customization and processing of background data. The batch parameters used for processing 88 RAW files are provided in *Figure 1—source data 4*. Two values – peak height and peak areas – were obtained for all peaks with an intensity >500 in each sample. This threshold was set to eliminate most of the background noise, based on empirical observations of raw chromatograms.

We further calculated different measures of diversity using the approach highlighted previously (*Martínez and Reyes-Valdés, 2008*; *Li et al., 2016*). Specifically, using peak intensity as a measure of count, we calculated Shannon Entropy (H) as follows:

$$Hj = -\sum_{i=1}^{m} Pij . \log 2(Pij)$$

where $P_{ij}$ indicates the relative frequency of the $i$th $m/z$ peak ($i = 1, 2, ..., m$) in the $j$th sample ($j = 1, 2, ...t$).

The average frequency $p_i$ of the $i$th $m/z$ peak among all samples was calculated as:

$$Pi = \frac{1}{t} \sum_{j=1}^{t} Pij$$

The specificity of the $i$th $m/z$ peak ($Si$) ws calculated as:

$$Si = \frac{1}{t} \sum_{j=1}^{t} (Pij/Pi). \log 2(Pij/Pi)$$

The specialization index of each sample $\delta j$ was measured for each $j$th sample as the average of the peak specificities using the following formula:

$$\delta j = \sum_{i=1}^{m} PijSi$$

## Purification and structure elucidation of acylsugars using NMR

For metabolite purification, aerial tissues of 28 Salpiglossis plants (aged 10 weeks) were extracted in 1.9 L of acetonitrile:isopropanol (AcN:IPA, v/v) for ~10 mins, and ~1 L of the extract was concentrated to dryness on a rotary evaporator and redissolved in 5 mL of AcN:IPA. Repeated injections from this extract were made onto a Thermo Scientific Acclaim 120 C18 HPLC column (4.6 × 150 mm, 5 μm particle size) with automated fraction collection. HPLC fractions were concentrated to dryness under vacuum centrifugation, reconstituted in AcN:IPA and combined according to metabolite purity as assessed by LC/MS. Samples were dried under $N_2$ gas and reconstituted in 250 or 300 μL of deuterated NMR solvent $CDCl_3$ (99.8 atom % D) and transferred to solvent-matched Shigemi tubes for analysis. $^1H$, $^{13}C$, $J$-resolved $^1H$, gCOSY, gHSQC, gHMBC and ROESY NMR experiments were performed at the Max T. Rogers NMR Facility at Michigan State University using a Bruker Avance 900 spectrometer equipped with a TCI triple resonance probe. All spectra were referenced to non-deuterated $CDCl_3$ solvent signals ($\delta_H$ = 7.26 and $\delta_C$ = 77.20 ppm).

## Extraction and sequencing of RNA

Total RNA was extracted from 4 to 5 week old (*S. nigrum, S. quitoense*) or 7–8 week old plants (*H. niger*, Salpiglossis) using Qiagen RNEasy kit (Qiagen, Valencia, California) with on-column DNA digestion. In addition to trichome RNA, total RNA was extracted from shaved stems of *S. nigrum* and Salpiglossis, and shaved petioles of *S. quitoense* and *H. niger*. The quality of extracted RNA was determined using Qubit (Thermo Fisher Scientific, USA) and Bioanalyzer (Agilent Technologies, Palo Alto, California). Total RNA from all 16 samples (4 species x 2 tissues x two biological replicates) was sequenced using Illumina HiSeq 2500 (Illumina, USA) in two lanes (8X multiplexing per lane). Libraries were prepared using the Illumina TruSeq Stranded mRNA Library preparation kit LT, sequencing carried out using Rapid SBS Reagents in a 2 × 100 bp paired end format, base calling done by Illumina Real Time Analysis (RTA) v1.18.61 and the output of RTA was demultiplexed and converted to FastQ format by Illumina Bcl2fastq v1.8.4.

## RNA-seq data analysis

The mRNA-seq reads were adapter-clipped and trimmed using Trimmomatic v0.32 using the parameters (*LEADING:20 TRAILING:20 SLIDINGWINDOW:4:20 MINLEN:50*). The quality-trimmed reads from all datasets of a species were assembled de novo into transcripts using Trinity v.20140413p1 after read normalization (max_cov = 50,KMER_SIZE = 25,max-pct-stdev=200,SS_lib_type = RF) (*Grabherr et al., 2011*). We tested three different kmer values (k = 25, 27, 31) and selected the best kmer value for each species based on contig N50 values, BLASTX hits to the *S. lycopersicum* annotated protein sequences and CEGMA results (*Parra et al., 2007*). A minimum kmer coverage of 2 was used to reduce the probability of erroneous or low abundance kmers being assembled into transcripts. After selecting the best assembly for each species, we obtained a list of transcripts differentially expressed between trichomes and stem/petiole for each species using RSEM/EBSeq (*Li and Dewey, 2011*; *Leng et al., 2013*) at an FDR threshold of p<0.05. The differential expression of five transcripts in *S. quitoense* and four transcripts in Salpiglossis was confirmed using semi-quantitative RT-PCR, along with the PDS as negative control (*Figure 2—figure supplement 3*).

## Prediction of protein sequences and orthologous group assignments

The protein sequences corresponding to the longest isoform of all expressed transcripts (read count >10 in at least one dataset in a given species) were obtained using TransDecoder (*Haas et al., 2013*) and GeneWise v2.1.20c (*Birney et al., 2004*). Only the protein sequences of transcripts with ≥10 reads as defined by RSEM were used for constructing orthologous groups using OrthoMCL v5 (*Li et al., 2003*). We defined orthologous relationships between *S. lycopersicum*, *S. nigrum*, *S. quitoense*, *H. niger*, *N. benthamiana*, Salpiglossis and *Coffea canephora* (outgroup) using an inflation index of 1.5.

## Gene ontology enrichment analyses

We transferred the tomato gene ontology assignments to the homologs from other species in the same orthologous group. GO enrichment analysis was performed using a custom R script, and enriched categories were obtained using Fisher Exact Test and correction for multiple testing based on Q-value (*Storey, 2002*).

## qRT-PCR

Primers to specific regions of the targeted transcript were designed with amplicons between 100–200 bp using Primer3 (*Untergasser et al., 2012*). The regions selected for amplification did not overlap with the region targeted in the VIGS analysis. Primer sets for Salpiglossis orthologs of PDS and elongation factor alpha (EF1a) were used as controls. We used 1 μg of total RNA from a single VIGS plant with an acylsugar phenotype to generate cDNA using the Thermo Fisher Superscript II RT kit. An initial amplification and visualization on a 1% agarose gel was performed to ensure that the primers yielded an amplicon with the predicted size and did not show visible levels of primer dimers. We first tested multiple primer sets per gene and selected primers within 85–115% efficiency range using a dilution series of cDNA from uninoculated plants. These primers were used for the final qRT-PCR reaction. The Ct values for the transcripts (on 1x template) were measured in triplicate, which were averaged for the analysis. Both uninoculated and empty vector controls were measured with all primer sets for ΔΔCt calculations.

## Confirmation of *Salpiglossis sinuata*

We confirmed the phylogenetic positions of Salpiglossis and *Hyoscyamus niger* using the chloroplast *ndhF* and *trnLF* marker based phylogenies (*Figure 2—figure supplement 1A,B*). Specifically, we amplified these regions using locus-specific primers, sequenced the amplicons and assembled the contigs using Muscle (*Edgar, 2004*). A neighbor joining tree including *ndhF* and *trnLF* sequences from NCBI Genbank was used to confirm the identity of the plant under investigation. Phylogenetic position of *S. nigrum* was confirmed based on BLASTN vs. all *S. nigrum* nucleotide sequences from NCBI. Several DNA barcodes (e.g. *trnLF* intergenic spacer, *atpFH* cds) showed 100% identity to *S. nigrum* RNA-seq transcripts over 100% of their lengths.

## Candidate gene amplification and enzyme assays

Enzyme assays were performed as previously described (*Fan et al., 2016b*) with the following modifications: Enzyme assays were performed in 30 μL reactions (3 μL enzyme +3 μL 1 mM acyl CoA +1 μL 10 mM (acylated) sucrose +23 μL 50 mM $NH_4Ac$ buffer pH 6.0) or 20 μL reactions with proportionately scaled components. Reactions that used an NMR-characterized substrate were performed with 0.2 μL substrate and 23.8 μL buffer. Reaction products were characterized using Waters Xevo G2-XS QToF LC/MS (Waters Corporation, USA) using previously described protocols (*Fan et al., 2016b*).

## Characterizing the SsASAT5 activity

Identification of SsASAT5 activity required a significant amount of the starting substrate, however, standard sequential reactions used to isolate SsASAT1,2,3 activities do not produce enough starting substrate for SsASAT5. Hence, we used S3:15(5,5,5) purified from a back-crossed inbred line (BIL6180, [*Ofner et al., 2016*]) whose NMR resolved structure shows R2, R3, R4 positions acylated by C5 chains (unpublished data) — same as Salpiglossis tri-acylsugars. SsASAT5 could use this substrate to acylate the furanose ring (*Figure 2—figure supplement 8A,B*), however, the tetra-

acylsugar thus produced is not further acetylated to penta-acylsugar, suggesting the C2 acylation by SsASAT5 is not the same as the C2 acylation on tetra-acylsugars in vivo. SsASAT5 also acetylated the acceptors S4:21(5,5,5,6) and S4:19(2,5,6,6) purified from Salpiglossis trichome extracts on the furanose ring to penta-acylsugars that co-migrated with the most abundant penta-acylsugars purified from the plant (*Figure 2C,D*). Finally, S5:23(2,5,5,5,6) and S5:21(2,2,5,6,6) purified from plant extracts were enzymatically acetylated to hexa-acylsugars, which co-migrated with two low abundance hexa-acylsugars from the plant (*Figure 2—figure supplement 8C,D*). Positive mode fragmentation patterns suggested that SsASAT5 possessed the capacity to acetylate both pyranose (weak) and furanose rings of the penta-acylsugars (*Figure 2—figure supplement 8E,F*).

## Virus induced gene silencing

Primers to amplify fragments of transcripts for VIGS were chosen to minimize probability of cross silencing other Salpiglossis transcripts due to sequence similarity or due to homopolymeric regions (*Supplementary files 2,3*). This was achieved using a custom Python script (*Moghe, 2017*; copy archived at https://github.com/elifesciences-publications/2017_Solanaceae), which divided the entire transcript of interest in silico into multiple overlapping 20nt fragments, performed a BLAST against the Salpiglossis transcriptome and tomato genome and flagged fragments with >95% identity and/ or >50% homopolymeric stretches. Contiguous transcript regions with >12 unflagged, high-quality fragments were manually inspected and considered for VIGS. 1–2 300 bp regions were cloned into the pTRV2-LIC vector (*Dong et al., 2007*) and transformed into *Agrobacterium tumefaciens* GV3101. Agro-infiltration of Salpiglossis plants was performed as described previously (*Velásquez et al., 2009*) using the prick inoculation method. Salpiglossis phytoene desaturase was used as the positive control for silencing. Empty pTRV2-LIC or pTRV2 vectors were used as negative controls. Each VIGS trial was done slightly differently while modifying the growth, transformation and maintenance conditions of this non-model species. The experimental details, including the optimal growth and VIGS conditions that give the fastest results, are noted in *Supplementary file 4*.

## Phylogenetic inference

All steps in the phylogenetic reconstruction were carried out using MEGA6 (*Tamura et al., 2013*). Amino acid sequences were aligned using Muscle with default parameters. Maximum likelihood and/ or neighbor joining (NJ) were used to generate phylogenetic trees. For maximum likelihood, the best evolutionary model (JTT[Jones-Taylor-Thornton]+G + I with five rate categories) was selected based on the Akaike Information Criterion after screening several models available in the MEGA6 software, while for NJ, the default JTT model was used. Support values were obtained using 1000 bootstrap replicates, however trees obtained using 100 bootstrap replicates also showed similar overall topologies. Trees were generated either using 'complete deletion' or 'partial deletion with maximum 30% gaps/missing data' options for tree reconstruction. Sequences < 350 aa (eg: Solyc10g079570) were excluded from this analysis. Trees were also generated using RAxML v8.0.6 (*Stamatakis, 2006*) with 1000 rapid bootstrap replicates, using the best models as specified by the Akaike Information Criterion in MEGA6.

## Similarity searches using BLAST

Similarity searches against the 1kp and NCBI nr databases were performed using TBLASTN, using SlASAT1, SlASAT3, SsASAT1 and SsASAT2 protein sequences as queries. For the 1kp database, TBLASTN was performed against all Asterid sequences, and the best non-Solanaceae sequences were analyzed using phylogenetic tree reconstruction (see below; *Figure 6—source data 1*). The BLAST at NCBI was performed against several specific groups of species, namely (i) orders Gentianales + Boraginales + Lamiales + Solanales-(family Solanaceae), (ii) Only Convolvulaceae, (iii) Solanaceae-(sub-family Solanoideae), (iv) Solanoideae-(genera Solanum + Capsicum), (v) Solanoideae-(genus Solanum), (vi) Solanum-(section Lycopersicon +species tuberosum). The top 10 hits from each search were manually curated and analyzed in a phylogenetic context with SlASATs and SsASATs (*Figure 6—figure supplements 2* and *3*). We used a similar approach to search the annotated peptide sequences in *C. canephora* (*Denoeud et al., 2014*) and *Ipomoea trifida* (*Hirakawa et al., 2015*) sequence databases. Sequences that provided additional insights into the evolution of the ASAT clade were integrated into the final gene tree shown in *Figure 6A*.

### Identification of Solanaceae-specific WGT event

We used the overall approach employed in Supplementary Note 5 of (*Bombarely et al., 2016*), where the authors obtained dS distributions between *S. lycopersicum* and Petunia homologs. We identified homologs using an all-vs-all protein BLAST, followed by selection of five top matches. The BLAST results and the genomic locations of the two gene sets were provided as input to MCScanX (*Wang et al., 2012*) with default parameters. The collinear blocks identified by MCScanX were used for calculation of dS values using the yn00 utility in PAMLv4.4 (*Yang, 2007*).

### Synteny analysis

Synteny between the Petunia, Capsicum and tomato genomes was determined as previously described using MCScanX (*Moghe et al., 2014*; *Wang et al., 2012*). Regions corresponding to PaA-SAT1 and its best matching homologs were used to make *Figure 7B*.

### Estimating the number of acylsugars

This and previous studies provide evidence for a number of acyl chains esterified to sucrose, classified into short (C2, C3, C4, iC5, aiC5, iC6, aiC6, C8), long (nC10, iC10, nC11, nC12) and non-aliphatic (malonyl). We only focused on aliphatic chains for this estimate. We typically see one long chain and 3–4 short chains on the sugar molecule across different Solanaceae species. In our calculation, we assumed that each acyl chain has the same probability of being incorporated into the acylsugar, with the core being a sucrose core. Under these assumptions, there are eight acyl chains that could be incorporated at three positions on a tetra-acylsugar and 12 acyl chains on the fourth position. This gives a theoretical estimate of 8*8*8*12 = 6144 acylsucroses that could be produced with the above acyl chains.

### Data release

All RNA-seq datasets are deposited in the NCBI Short Read Archive under the BioProject PRJNA263038. Coding sequences of ASATs are provided in *Supplementary file 2* and have been deposited in GenBank (KY978746-KY978750). RNA-seq transcripts and orthologous group membership information has been uploaded to Dryad (provisional DOI: 10.5061/dryad.t7r64).

## Acknowledgements

We thank Dr. Rachel Meyers, Dr. Amy Litt and the New York Botanical Gardens as well as Dr. Frank Telewski at the MSU Beal Botanical Gardens for permission and assistance in sample collection. We also thank the support staff at the Research Technology Support Facility at MSU for transcriptome sequencing and mass spectrometry assistance, and the Max T Rogers NMR Facility for assistance with NMR data generation. We acknowledge Daniel Lybrand for cloning the *Solanum nigrum ASAT1* gene, and Dr. Anthony Schilmiller and Dr. Pengxiang Fan for assisting with sample collection at NYBG. We also acknowledge the valuable feedback received from members of the Last lab, Dr. Cornelius Barry and Dr. Eran Pichersky in the research design of this manuscript. We especially thank Dr. Barry and Dr. Swathi Nadakuduti for sharing their unpublished data. This work was funded by National Science Foundation grants IOS–1025636 and IOS-PGRP-1546617 to RLL and ADJ and National Institute of General Medical Sciences of the National Institutes of Health graduate training grant no. T32–GM110523 to BL ADJ acknowledges support from the USDA National Institute of Food and Agriculture, Hatch project MICL-02143.

## Additional information

### Funding

| Funder | Grant reference number | Author |
| --- | --- | --- |
| National Institutes of Health | T32-GM110523 | Bryan J Leong<br>Robert L Last |
| National Science Foundation | IOS-1025636 | A Daniel Jones<br>Robert L Last |

| U.S. Department of Agriculture | MICL-02143 | A Daniel Jones |
| --- | --- | --- |
| National Science Foundation | IOS-PGRP-1546617 | A Daniel Jones<br>Robert L Last |

The funders had no role in study design, data collection and interpretation, or the decision to submit the work for publication.

### Author contributions

Gaurav D Moghe, Conceptualization, Resources, Data curation, Software, Formal analysis, Supervision, Validation, Investigation, Methodology, Writing—original draft, Project administration; Bryan J Leong, Investigation, Writing—review and editing; Steven M Hurney, Formal analysis, Investigation, Methodology; A Daniel Jones, Conceptualization, Formal analysis, Funding acquisition, Investigation, Methodology, Writing—review and editing; Robert L Last, Conceptualization, Supervision, Funding acquisition, Writing—original draft, Project administration, Writing—review and editing

### Author ORCIDs

Gaurav D Moghe http://orcid.org/0000-0002-8761-064X
Bryan J Leong http://orcid.org/0000-0003-4042-1160
A Daniel Jones http://orcid.org/0000-0002-7408-6690
Robert L Last http://orcid.org/0000-0001-6974-9587

### Decision letter and Author response

Decision letter https://doi.org/10.7554/eLife.28468.057
Author response https://doi.org/10.7554/eLife.28468.058

## Additional files

### Supplementary files

• Supplementary file 1. GO categories enriched among trichome-high genes in different Solanaceae species. The enrichment columns show whether the GO is enriched in trichome high transcripts in this species, based on Fisher Exact Test 1: yes, 0: no. In the '# of genes' column, 0 = GO category not enriched among trichome high genes in this species.
DOI: https://doi.org/10.7554/eLife.28468.047

• Supplementary file 2. Sequences identified in this study. The transcript identifier as per the assembler Trinity nomenclature is noted in the part after the bar (|). Green and yellow highlighted sequences are the two regions targeted for VIGS in Salpiglossis.
DOI: https://doi.org/10.7554/eLife.28468.048

• Supplementary file 3. Primer sequences used in this study.
DOI: https://doi.org/10.7554/eLife.28468.049

• Supplementary file 4. Experimental conditions for VIGS experiments.
DOI: https://doi.org/10.7554/eLife.28468.050

• Supplementary file 5. MEGA6 aligned file in the *. mas format. These sequences were used to make *Figure 6A*.
DOI: https://doi.org/10.7554/eLife.28468.051

### Major datasets

The following datasets were generated:

| Author(s) | Year | Dataset title | Dataset URL | Database, license, and accessibility information |
| --- | --- | --- | --- | --- |
| Moghe GD | 2015 | RNA seq reads from Solanaceae Spp. | https://www.ncbi.nlm.nih.gov/bioproject/263038 | Publicly available at the NCBI BioProject (accession no: PRJNA263038) |

| Moghe GD | 2017 | Data from: Multi-omic analysis of a hyper-diverse plant metabolic pathway reveals evolutionary routes to biological innovation | http://dx.doi.org/10.5061/dryad.t7r64 | Available at Dryad Digital Repository under a CC0 Public Domain Dedication |
|---|---|---|---|---|

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
