## [Decision Letter]

Thank you for submitting your article "Multi-omic analysis of a hyper-diverse plant metabolic pathway reveals evolutionary routes to biological innovation" for consideration by *eLife*. Your article has been favorably evaluated by Detlef Weigel (Senior Editor) and three reviewers, one of whom is a member of our Board of Reviewing Editors. The following individual involved in review of your submission has agreed to reveal their identity: Alisdair R Fernie (Reviewer #3).

The reviewers have discussed the reviews with one another and the Reviewing Editor has drafted this decision to help you prepare a revised submission.

Summary:

The authors provide a very strong biochemical and transcriptional analysis of the evolution of enzymes within acylsugar metabolism across a wide phylogeny. This work helps to contribute to our rapidly developing understanding of evolution in defense metabolism.

Essential revisions:

The reviewers all agree that there were three key points that should be addressed to strengthen the manuscript.

1) The question of whole genome duplications in this process should be assessed as best as possible given the available transcriptomic and/or genomic sequences. Especially given the known WGD in the Solanaceae.

2) Metabolite diversity should be directly tested using any of the available statistical approaches used in ecology and evolution to test variation in diversity.

3) The writing needs to be tighter on what has been and not been shown in the literature to date, what this manuscript shows and what this manuscript contributes to our broader understanding of secondary metabolism.

*Reviewer #1:*

This is a potentially interesting manuscript about the evolution of structural diversity in acyl-sugars within the solanaceae. The strengths of the manuscript are the chemistry and the analysis of the acyl-sugar involved genes. There are some weaknesses that should be addressed. The first is the vague use of diversity for multiple meanings throughout the manuscript as described below. The second is that there is no evidence that this diversity has in planta mechanistic function. This could have been solved by testing the SsASAT lines for biotic activity but this was not done.

The authors work on acylsugars that in general have been linked to anti-herbivore activities but as of yet, there has been little evidence showing that the diversity in acyl sugar structures leads to different in planta biotic activities as has been shown directly for glucosinolates in the Brassicales and DIMBOA compounds in Maize. In the third paragraph of the Introduction, the authors say that there is evidence that the diversity is functionally important. But these citations, Puterka and Leckie, only looked at purified acyl sugars using in vitro studies and do not really allow the claim that this diversity matters in a real plant. This generates a central weakness in this study that there is no in planta evidence that this diversity really matters using single gene manipulations. There is a similar level of diversity in the glycosylated flavonols throughout the plant kingdom where the hexose pattern on a single compound like Kaempferol leads to a massive structural diversity. Yet this structural diversity has yet to be given a biological reason to exist. A simple solution to this would have been testing the transgenic lines with simpler acylsugar patterns for biotic resistance. As this was not done, it seems a lot to ask to include in an already large paper. A better solution is to give a more realistic accounting that the diversity of structure in this pathway has little to no evidence of activity and this needs to be tested in the future. The present Introduction and Discussion left this reader thinking that there was proven in planta activity for this diversity but a reading of the citations showed that this was not the case. As the strength is the family level analysis of gene level creation of structural diversity, there is no reason to hide this weakness. All that happens is that the follow-up study showing the biological role looks less impressive.

For Figures like Figure 3 and others, there are direct measures like Shannon Entropy that can be used to directly quantify structural diversity and use that number in statistical analysis. This would allow the authors to directly show that the acyl sugar pattern is becoming simpler in the SsASAT2 lines. This is a classic tool to look at ecosystem diversity in ecology and has a long history. If the claim is truly about diversity, then it would seem to be best to directly test diversity.

There is a recurring confusion about the term diversity within this manuscript that is a central weakness to this reader. This has several foundations. One is that the authors interchange diversity across multiple levels (within individual, within species and between species) without ever clarifying or even distinguishing these three. For example, the authors say that Prasad et al. 2012 shows that glucosinolate diversity influences fitness in the wild. Yet this paper is not between species or within individual diversity but about two different alleles that simply alter the structure between two individuals (within species) and showing that one allele is better than another. This then is directly compared to the Macel citation which is not about within species but between species comparisons. And finally, this is compared to Weinhold and Baldwin, 2011; Leckie et al., 482 2016; and Luu et al., 2017 papers that are about the possible mechanistic function of O-acyl sugars in general without delving into the role of diversity directly. As such, these three citation sets are not directly comparable as they do not discuss the same diversity.

Furthering this confusion is that diversity is interchangeably used to mean the blend of structures possible at all levels (individual, species and family) while simultaneously meaning the differences between these groupings. Yet these are very different meanings with the blend of structures representing potential and the individual composition meaning actuality. For example, the Prasad citation talks about two different groups of structures in glucosinolates and the plant has to choose between one or the other and cannot have both in that particular situation. As such, it is essential to qualify all the different meanings of diversity in a paper that discusses chemical structures, genetic variation within and between species and even between family.

As a side on the topic of diversity and fitness, there are papers by Kerwin and Zust on glucosinolates that show that diversity of structures within species is beneficial to the species wherein one allele is optimal in one environment and an alternative is optimal in another using direct assays of individual biosynthetic genes effect on field fitness. These specifically address the fluctuating selection/bet-hedging model of secondary metabolism whereby the genetic diversity to chemical diversity link provides a unique solution to fluctuating biotrophic attack.

In the second paragraph of the subsection “Conclusion”, the authors state "Based on our findings, the acylsugar phenotype across the Solanaceae is "hyper-diverse", meaning there is large diversity within and between closely related species". What is hyper-diverse? This is not a quantitative nor qualitative definition as there is no comparison to other species and other compound classes. The authors then compare this to other species/compounds, but this is not an easy comparison as most of these other compound classes have not been targeted with LC-MS platforms with the sensitivity as used in this manuscript. A fundamental question that this raises is what is the benefit of saying hyper-diverse beyond some inherent claim that our pathway is more structurally (key) diverse than yours.

The Moghe and Last citation in use for glucosinolates is presently out of date as there have been recent papers showing the role of whole genome duplications and tandem gene duplications that enabled the evolution of the whole pathway beyond the single isopropylmalate synthase event. Part of the reason for this is that in the glucosinolate pathway, the duplications allowed the old structures to be kept when making the new ones but in the acylsugar pathway, this does not seem to be the case. As this manuscript is working to discuss a similar level of genomic/family level evolution that was worked out last year in Glucosinolates, the authors should directly cite the relevant primary literature. Especially as there is no limit on citations for *eLife*, it makes sense to give credit to the people who found all of the evidence in the second paragraph of the Introduction. Equally, isn't Ohno, the book that came up with the idea, a better citation for sub- and neo-functionalization?

*Reviewer #2:*

The authors claim that "this study provides mechanistic insights into the emergence of a plant chemical novelty, and offers a template for investigating the 300,000 non-model plant species that remain underexplored." While I do not doubt the biochemical experiments, the template provided misses the mark on evolutionary analyses, and does not take advantage of any new network methods. Some brief comments:

1) Experimental Design. A reference phylogeny is given in Figure 1 for the Solanaceae; however, the manuscript wants to address the evolution of the acylsugar biosynthesis pathway across the orders Solanales, Lamiales, and Boraginales also. If this was the case, why not use an expanded phylogeny to guide collecting representatives from across these orders and within the Solanaceae (with matched tissue for both metabolomics and transcriptomics)?

2) Phylogenetics. Given that the authors gathered transcriptome data, why not create transcriptome phylogenies based on thousands of nuclear genes to confirm species identity and investigate the gene families of interest at the same time? These could have been generated initially, but otherwise it is easy enough to mine the transcriptomes from the OneKP project. Futhermore, BLAST is only one way to use transcriptome data which can now be anchored to related genomes. This is now a well-established methodology in the plant genomics community (see Brockington et al. 2015 cited by the authors, but also many other studies, including ones that specifically look at metabolism in both model and non-model systems).

Also, the neighbor joining analyses used Figure 1—figure supplement 1 and Figure 2—figure supplement 1 were abandoned in the mid-1990s for more sophisticated likelihood and Bayesian analyses.

3) Evolution of duplicates. The authors make claims about evolution and duplication; however, they do not use the genomic resources to phylogenetically evaluate whether the duplicates being considered are from whole genome duplication (WGD or polyploidy) vs. small scale duplication (SSD). Given that Solanaceae have a WGD, it would seem important to investigate the acylsugar biosynthesis pathway in this context. Several papers not cited on the origins of metabolic pathways have confronted this link directly (for example; Edger et al. 2015 PNAS 112: 8362; Zhou et al. 2016 *eLife*, 5: e19531). Without such analyses, it is not clear what claims can be made about the nature of gene and genome duplication.

4) Reconstructing pathways using synteny and/or network approaches. Given the genomic resources of the Solanales and related orders, the acylsugar biosynthetic pathway should be evaluated using synteny (and there are online tools to do so, such as COGE). Similarly, the authors miss an opportunity to engage the debate about the origins of metabolic pathways and tools to investigate this (for example, Wisecaver et al. 2017. Plant Cell "A global co- expression approach for connecting genes to specialized metabolic pathways in plants."). The novelty of the manuscript would be elevated if newer methods were utilized.

5) Novelty. Not being familiar with the author's previous work, it would be helpful to state up front what is known from their recent work (Ning et al. 2015, Fan et al. 2016a, b, Schilmiller et al. 2016, etc.) and what is explicitly new here. As far as this reviewer can tell, the main methodological novelty of the paper is the development of VIGS for the non-model genus Salpiglossis.

*Reviewer #3:*

This is a highly interesting and well carried out study that documents a logical extension to previous work in the Last lab albeit one carried out on a massive scale. I would like to start my review with the admission that large sectors of the evolutionary analysis are beyond my ability to evaluate – the authors conclusions seem reasonable to me however I cannot comment deeper on these aspects. The metabolite analysis which I can comment in detail on is exemplary and the transcriptome analysis also. One thing that I feel is lacking however is the quantitative data for the metabolites I think this should be available and probably be subjected to a deeper statistical characterization than they currently are. I agree this is not needed to support the current manuscript but believe that it would render the manuscript a more powerful reference resource.

My other comments concern the writing style which, on occasion, I feel is exaggerated. This starts with the first word of the title – I admit I am not a fan of the "word" omics but feel that multi-omics should probably be toned down for this study. Many other sentences claim the generality of the study to metabolism across the plant kingdom. I am not sure these are quite substantiated and therefore suggest that they modify such claims to make them more accurate to what they actually show or clearly indicate when they are speculating. In line with this I would have liked to see a slightly longer perspective as to what doors such broad studies offer i.e. do they provide a platform to assess the drivers of the evolution of chemical diversity and hence the fitness of plants acquiring new chemical tools to their arsenal. Whilst hints as to this are included in the text it would be nice if the authors could distill this in a more clear fashion.

---

## [Author Response]

*Essential revisions:*

*The reviewers all agree that there were three key points that should be addressed to strengthen the manuscript.*

*1) The question of whole genome duplications in this process should be assessed as best as possible given the available transcriptomic and/or genomic sequences. Especially given the known WGD in the Solanaceae.*

In the previous version of our manuscript, we discussed two duplication events:

i) A duplication event that gave rise to the ASAT clade (blue subclade – Figure 6): This event occurred before Solanaceae-Convolvulaceae divergence, in contrast to the WGT event, which occurred after the divergence between the families (Bombarely et al., Nature Plants, 2016). We have now clarified this in the fourth paragraph of the subsection “The evolutionary origins of acylsugar biosynthesis”.

ii) Duplication events that gave rise to the multiple ASATs: We have now performed additional analyses that suggest the many duplications that gave rise to the major BAHD clades in the blue subclade also occurred prior to the Solanaceae-specific WGT event (subsection “The evolutionary origins of acylsugar biosynthesis”, fifth paragraph; Figure 6). We used the same framework as described in Supplementary Note 5 of Bombarely et al., Nature Plants, 2016, using within- and between-genome syntenic blocks and synonymous substitution rate to identify the timing for the Solanaceae-specific WGT event and the order of duplications in the blue subclade.

*2) Metabolite diversity should be directly tested using any of the available statistical approaches used in ecology and evolution to test variation in diversity.*

We have now used the approach described in Li et al., PNAS, 2016 to calculate Shannon Entropy and degree of specialization for different acylsugar profiles. These results are now shows as Figure 1 and Figure 1—figure supplement 5. The methods are described in Materials and methods section (subsection “Calculation of Shannon Entropy”), and the source data used for the calculation are also provided (Figure 1—source data 4, Figure 1—source data 5). We found that Salpiglossis and *Nicotiana alata* have the highest entropy of surface metabolite profiles and that these profiles are generally specific to the plant species and tissues tested.

*3) The writing needs to be tighter on what has been and not been shown in the literature to date, what this manuscript shows and what this manuscript contributes to our broader understanding of secondary metabolism.*

We have now discussed results from previous studies in detail in the Introduction (third paragraph). We also expanded the Discussion regarding evolution of specialized metabolic pathways, the advantages of using integrative approaches such as the one in this study and the need for development of approaches to study metabolic diversity in non-model organisms.

*Reviewer #1:*

*This is a potentially interesting manuscript about the evolution of structural diversity in acyl-sugars within the solanaceae. The strengths of the manuscript are the chemistry and the analysis of the acyl-sugar involved genes. There are some weaknesses that should be addressed. The first is the vague use of diversity for multiple meanings throughout the manuscript as described below.*

We thank the reviewer for considering our study positively and for providing a very useful critique. Hopefully, this new version of the manuscript has resolved most of this reviewer’s concerns. We have responded to the diversity issue in our fourth response below.

*The second is that there is no evidence that this diversity has in planta mechanistic function. This could have been solved by testing the SsASAT lines for biotic activity but this was not done.*

*The authors work on acylsugars that in general have been linked to anti-herbivore activities but as of yet, there has been little evidence showing that the diversity in acyl sugar structures leads to different in planta biotic activities as has been shown directly for glucosinolates in the Brassicales and DIMBOA compounds in Maize. In the third paragraph of the Introduction, the authors say that there is evidence that the diversity is functionally important. But these citations, Puterka and Leckie, only looked at purified acyl sugars using* in vitro *studies and do not really allow the claim that this diversity matters in a real plant. This generates a central weakness in this study that there is no in planta evidence that this diversity really matters using single gene manipulations. There is a similar level of diversity in the glycosylated flavonols throughout the plant kingdom where the hexose pattern on a single compound like Kaempferol leads to a massive structural diversity. Yet this structural diversity has yet to be given a biological reason to exist. A simple solution to this would have been testing the transgenic lines with simpler acylsugar patterns for biotic resistance. As this was not done, it seems a lot to ask to include in an already large paper. A better solution is to give a more realistic accounting that the diversity of structure in this pathway has little to no evidence of activity and this needs to be tested in the future. The present Introduction and Discussion left this reader thinking that there was proven in planta activity for this diversity but a reading of the citations showed that this was not the case. As the strength is the family level analysis of gene level creation of structural diversity, there is no reason to hide this weakness. All that happens is that the follow-up study showing the biological role looks less impressive.*

As the reviewer rightly pointed out, the ecological role of acylsugar diversity is still a largely unresolved question. We have now made it clearer in the manuscript in the Introduction section (third paragraph) and in the Conclusions section (third paragraph).

In our opinions, the most compelling study discussing the ecological roles of acylsugars comes from the Baldwin lab (Weinhold and Baldwin, PNAS, 2011) and we cited this study in the initial submission. We have now expanded upon its results for describing the ecological importance of acyl chain diversity in acylsugars.

The VIGS lines generated in our study possess only a transient knockdown of the ASAT enzymes. Hence, we did not consider them optimal for rigorously testing the importance of acylsugar diversity. Additional experiments that use stable knockdowns/knockouts and gene editing (e.g. RNAi, CRISPR-Cas9) would be useful for clearly addressing this question and would require development of transformation in Salpiglossis and other species.

*For Figures like Figure 3 and others, there are direct measures like Shannon Entropy that can be used to directly quantify structural diversity and use that number in statistical analysis. This would allow the authors to directly show that the acyl sugar pattern is becoming simpler in the SsASAT2 lines. This is a classic tool to look at ecosystem diversity in ecology and has a long history. If the claim is truly about diversity, then it would seem to be best to directly test diversity.*

Thank you for this suggestion. We calculated Shannon Entropy of acylsugar-like peaks in acylsugar producing species, where we originally made claims about diversity. The results of this analysis are described in the fifth paragraph of the subsection “Diversity of acylsugar profiles across the Solanaceae”, in Figure 1 and Figure 1—figure supplement 5. The approach is described in detail in the Materials and methods section. We found that Salpiglossis and *Nicotiana alata* have the highest entropy of surface metabolite profiles and that the profiles are generally very specific to the plant and tissues sampled.

With regards to diversity in the VIGS lines, our focus was primarily total acylsugar levels (SsASAT1, SsASAT2) and specific types of acylsugars produced (SsASAT3, SsASAT5) in the knockdowns that may inform us about the likely position of the perturbed enzyme in the biosynthetic pathway. We previously performed detailed analyses of these two aspects in Figure 3, Figure 4 and the associated figure supplements, using Kolmogorov-Smirnov tests to identify and describe specific, significant differences.

*There is a recurring confusion about the term diversity within this manuscript that is a central weakness to this reader. This has several foundations. One is that the authors interchange diversity across multiple levels (within individual, within species and between species) without ever clarifying or even distinguishing these three. For example, the authors say that Prasad et al. 2012 shows that glucosinolate diversity influences fitness in the wild. Yet this paper is not between species or within individual diversity but about two different alleles that simply alter the structure between two individuals (within species) and showing that one allele is better than another. This then is directly compared to the Macel citation which is not about within species but between species comparisons. And finally, this is compared to Weinhold and Baldwin, 2011; Leckie et al., 482 2016; and Luu et al., 2017 papers that are about the possible mechanistic function of O-acyl sugars in general without delving into the role of diversity directly. As such, these three citation sets are not directly comparable as they do not discuss the same diversity.*

*Furthering this confusion is that diversity is interchangeably used to mean the blend of structures possible at all levels (individual, species and family) while simultaneously meaning the differences between these groupings. Yet these are very different meanings with the blend of structures representing potential and the individual composition meaning actuality. For example, the Prasad citation talks about two different groups of structures in glucosinolates and the plant has to choose between one or the other and cannot have both in that particular situation. As such, it is essential to qualify all the different meanings of diversity in a paper that discusses chemical structures, genetic variation within and between species and even between family.*

In response to these important concerns, we went through the entire manuscript and checked the context of the word “diversity”. We attempt to use this word only to refer to diversity of chemical structures (e.g. diversity of acylsugars, sugar cores, acyl chains, glucosinolate structures). We have also attempted, to the best of our abilities, to make the taxonomic level clear at multiple places where diversity is discussed, especially in the Conclusions section (third paragraph). As noted below (in our sixth response), we also re-worded the section on acylsugar profiles being hyper-diverse.

*As a side on the topic of diversity and fitness, there are papers by Kerwin and Zust on glucosinolates that show that diversity of structures within species is beneficial to the species wherein one allele is optimal in one environment and an alternative is optimal in another using direct assays of individual biosynthetic genes effect on field fitness. These specifically address the fluctuating selection/bet-hedging model of secondary metabolism whereby the genetic diversity to chemical diversity link provides a unique solution to fluctuating biotrophic attack.*

Thanks for suggesting the Kerwin and Zust papers. We now include these points in the third paragraph of the subsection “Conclusions”.

*In the second paragraph of the subsection “Conclusion”, the authors state "Based on our findings, the acylsugar phenotype across the Solanaceae is "hyper-diverse", meaning there is large diversity within and between closely related species". What is hyper-diverse? This is not a quantitative nor qualitative definition as there is no comparison to other species and other compound classes. The authors then compare this to other species/compounds, but this is not an easy comparison as most of these other compound classes have not been targeted with LC-MS platforms with the sensitivity as used in this manuscript. A fundamental question that this raises is what is the benefit of saying hyper-diverse beyond some inherent claim that our pathway is more structurally (key) diverse than yours.*

We replaced the word hyper-diverse in the title, Abstract and text with a more accurate description of the phenotype. Specifically, we use the words “large diversity” or “highly diverse”. We removed the comparisons between the different metabolite classes and now simply say that the acylsugar diversity is reminiscent of metabolite diversity in other lineage-specific metabolite classes.

*The Moghe and Last citation in use for glucosinolates is presently out of date as there have been recent papers showing the role of whole genome duplications and tandem gene duplications that enabled the evolution of the whole pathway beyond the single isopropylmalate synthase event. Part of the reason for this is that in the glucosinolate pathway, the duplications allowed the old structures to be kept when making the new ones but in the acylsugar pathway, this does not seem to be the case. As this manuscript is working to discuss a similar level of genomic/family level evolution that was worked out last year in Glucosinolates, the authors should directly cite the relevant primary literature. Especially as there is no limit on citations for eLife, it makes sense to give credit to the people who found all of the evidence in the second paragraph of the Introduction. Equally, isn't Ohno, the book that came up with the idea, a better citation for sub- and neo-functionalization?*

We added relevant citations at the statements noted by the reviewer. We also cite Ohno 1970 (sub-functionalization) and Force, 1999 (neo-functionalization) in the second paragraph of the Introduction.

*Reviewer #2:*

*The authors claim that "this study provides mechanistic insights into the emergence of a plant chemical novelty, and offers a template for investigating the 300,000 non-model plant species that remain underexplored." While I do not doubt the biochemical experiments, the template provided misses the mark on evolutionary analyses, and does not take advantage of any new network methods. Some brief comments:*

*1) Experimental Design. A reference phylogeny is given in Figure 1 for the Solanaceae; however, the manuscript wants to address the evolution of the acylsugar biosynthesis pathway across the orders Solanales, Lamiales, and Boraginales also. If this was the case, why not use an expanded phylogeny to guide collecting representatives from across these orders and within the Solanaceae (with matched tissue for both metabolomics and transcriptomics)?*

In response to the suggestion we sampled additional plants in the Lamiales, Boraginales and Gentianales orders. Specifically, we performed extraction of leaf surface metabolites from 11 species in these orders. Our results are consistent with published results that surface acylsugars are restricted to the Solanaceae. It is also consistent with our finding that the duplication event that gave rise to the ASAT clade occurred after the node that separates Solanales and Lamiales/Boraginales orders. Results of this analysis are presented in Figure 6—figure supplement 4.

In our study, we used metabolite profiling as a way to screen for plants to be used for RNA-seq. We selected plants in the Solanaceae family – the family of most interest to us – that were phylogenetically dispersed and showed interesting acylsugar profiles. We originally did not sample families beyond Solanaceae-Convolvulaceae because there was no evidence in literature of acylsugars beyond Solanaceae, and because of absence of any ASAT homologs in BLAST searches beyond the family.

*2) Phylogenetics. Given that the authors gathered transcriptome data, why not create transcriptome phylogenies based on thousands of nuclear genes to confirm species identity and investigate the gene families of interest at the same time? These could have been generated initially, but otherwise it is easy enough to mine the transcriptomes from the OneKP project. Futhermore, BLAST is only one way to use transcriptome data which can now be anchored to related genomes. This is now a well-established methodology in the plant genomics community (see Brockington et al. 2015 cited by the authors, but also many other studies, including ones that specifically look at metabolism in both model and non-model systems).*

The reviewer has proposed an analysis that can be very useful to the field, but the original focus of our studies – for both scientific and budgetary reasons – was Solanaceae, given absence of evidence for acylsugars in non-Solanaceae species.

With regards to the OneKP database, we indeed searched the transcriptomes of all Solanales, Lamiales, Boraginales and Gentianales species in the OneKP project as mentioned in the Materials and methods (subsection: “Similarity searches using BLAST”), however, we did not find any close ASAT homologs through this search. The results of this BLAST are now provided as Figure 6—source data 1.

Furthermore, we have used multiple genome sequences in the relevant lineages, including *Coffea canephora* (Gentianales – used for making orthologous groups), *Mimulus guttatus* (Lamiales – used in BLAST searches for identifying ASAT homologs), *Ipomoea trifida* (Convolvulaceae – used in BLAST searches for identifying ASAT homologs) as well as *Petunia axillaris, Capsicum annuum, Solanum phureja* (Solanaceae – used for studying genome synteny using MCScanX). These analyses are described in the Materials and methods and shown in Figure 6 and Figure 7 and the associated supplemental figures and source data.

*Also, the neighbor joining analyses used Figure 1—figure supplement 1 and Figure 2—figure supplement 1 were abandoned in the mid-1990s for more sophisticated likelihood and Bayesian analyses.*

We have now re-made the figures using Maximum Likelihood.

*3) Evolution of duplicates. The authors make claims about evolution and duplication; however, they do not use the genomic resources to phylogenetically evaluate whether the duplicates being considered are from whole genome duplication (WGD or polyploidy) vs. small scale duplication (SSD). Given that Solanaceae have a WGD, it would seem important to investigate the acylsugar biosynthesis pathway in this context. Several papers not cited on the origins of metabolic pathways have confronted this link directly (for example; Edger et al. 2015 PNAS 112: 8362; Zhou et al. 2016 eLife, 5: e19531). Without such analyses, it is not clear what claims can be made about the nature of gene and genome duplication.*

We performed additional analysis to determine the role of whole genome duplication and tandem duplications in the evolution of acylsugar biosynthesis, as described previously. Results of this analysis are described in the fifth paragraph of the subsection “The evolutionary origins of acylsugar biosynthesis”, and in Figure 6. We have also cited the two suggested papers, which highlight the role of WGD and tandem duplications in evolution of plant defense metabolism.

*4) Reconstructing pathways using synteny and/or network approaches. Given the genomic resources of the Solanales and related orders, the acylsugar biosynthetic pathway should be evaluated using synteny (and there are online tools to do so, such as COGE). Similarly, the authors miss an opportunity to engage the debate about the origins of metabolic pathways and tools to investigate this (for example, Wisecaver et al. 2017. Plant Cell "A global co- expression approach for connecting genes to specialized metabolic pathways in plants."). The novelty of the manuscript would be elevated if newer methods were utilized.*

We performed MCScanX analysis among the sequenced genomes in the Solanaceae family, as shown in Figure 7. The approach is described in the Materials and methods section. In this new version, we also performed additional genome-synteny based analyses, as described above, to determine the involvement of WGT in the evolution of acylsugar biosynthesis. We did not include non-Solanaceae species in our synteny analyses because no ASAT homologs could be detected in those species.

As per the reviewer’s suggestion, we expanded our discussion on the evolution of metabolic pathways, and how the multi-omic approach as described in this study and other studies (e.g. Edger et al., PNAS, 2015; Zhou et al., *eLife*, 2016) can be used to understand the evolution of biological complexity (subsection “Conclusions”, fourth and fifth paragraphs).

*5) Novelty. Not being familiar with the author's previous work, it would be helpful to state up front what is known from their recent work (Ning et al. 2015, Fan et al. 2016a, b, Schilmiller et al. 2016, etc.) and what is explicitly new here. As far as this reviewer can tell, the main methodological novelty of the paper is the development of VIGS for the non-model genus Salpiglossis.*

Previous work from our lab has focused on identifying the acylsugar biosynthetic enzymes in cultivated tomato and understanding their evolution in two closely related species *S. habrochaites* and *S. pennellii* (~5 million years since speciation). In this study, we (i) discovered biosynthetic enzymes in Salpiglossis, *Hyoscyamus niger* and *Solanum nigrum* and (ii) expanded our understanding of acylsugar pathway evolution up to 100 million years of plant evolution. In the process, we discovered interesting metabolite profiles and evolutionary dynamics that occurred within the pathway, which serve as foundations for future research on the topic.

The reviewer is right in stating that Salpiglossis VIGS is a methodological novelty of this paper. In addition, we believe that the scope of the study, the integrative approach and the biological insights derived are also novel. For example, our manuscript explores the evolution of the entire pathway in contrast to most other studies that typically explore only one enzyme central to the pathway. We also attempted to emphasize the importance of and the relative ease with which an integrative approach such as ours (and other studies cited above) can be deployed in under-studied plant lineages, for understanding plant metabolic diversity. We emphasized these points in the “Conclusions” subsection.

*Reviewer #3:*

*This is a highly interesting and well carried out study that documents a logical extension to previous work in the Last lab albeit one carried out on a massive scale. I would like to start my review with the admission that large sectors of the evolutionary analysis are beyond my ability to evaluate – the authors conclusions seem reasonable to me however I cannot comment deeper on these aspects. The metabolite analysis which I can comment in detail on is exemplary and the transcriptome analysis also. One thing that I feel is lacking however is the quantitative data for the metabolites I think this should be available and probably be subjected to a deeper statistical characterization than they currently are. I agree this is not needed to support the current manuscript but believe that it would render the manuscript a more powerful reference resource.*

We thank Dr. Fernie for his encouraging words and suggestions. To address his concerns regarding quantitative data for metabolites, we have now performed an analysis of Shannon entropy to depict the diversity of acylsugars in the Solanaceae using the approach described previously (Li et al., PNAS, 2016).

*My other comments concern the writing style which, on occasion, I feel is exaggerated. This starts with the first word of the title – I admit I am not a fan of the "word" omics but feel that multi-omics should probably be toned down for this study. Many other sentences claim the generality of the study to metabolism across the plant kingdom. I am not sure these are quite substantiated and therefore suggest that they modify such claims to make them more accurate to what they actually show or clearly indicate when they are speculating.*

As per the reviewer’s suggestions, we have edited the title to remove the words “Multi-omics” and “hyper-diverse”. We also significantly re-worded our discussion on acylsugar phenotype being hyper-diverse. We have also reviewed our manuscript to state our findings more modestly.

*In line with this I would have liked to see a slightly longer perspective as to what doors such broad studies offer i.e. do they provide a platform to assess the drivers of the evolution of chemical diversity and hence the fitness of plants acquiring new chemical tools to their arsenal. Whilst hints as to this are included in the text it would be nice if the authors could distill this in a more clear fashion.*

We expanded our discussion on the broader implications of this approach (subsection “Conclusions”, last three paragraphs). Specifically, we discuss the utility of this integrative approach in discovering specialized metabolic pathways in non-model species, and understanding how the constituent enzymes have evolved. Understanding the principles that govern enzyme activity change in nature can also aid the cause of synthetic biology for e.g. by providing novel cassettes for production of natural products, as demonstrated for labdane-type diterpenoids (Ignea et al., Metab Eng, 2015).